# Do Sparse Autoencoders Identify Reasoning Features in Language Models?

George Ma [1]   Zhongyuan Liang [1 2]   Irene Y. Chen [1 2]   Somayeh Sojoudi [1]

## Abstract

We study how reliably sparse autoencoders (SAEs) support claims about reasoning-related internal features in large language models. We first give a stylized analysis showing that sparsity-regularized decoding can preferentially retain stable low-dimensional correlates while suppressing high-dimensional within-behavior variation, motivating the possibility that contrastively selected "reasoning" features may concentrate on cue-like structure when such cues are coupled with reasoning traces. Building on this perspective, we propose a falsification-based evaluation framework that combines causal token injection with LLM-guided counterexample construction. Across 22 configurations spanning multiple model families, layers, and reasoning datasets, we find that many contrastively selected candidates are highly sensitive to token-level interventions, with 45%–90% activating after injecting only a few associated tokens into non-reasoning text. For the remaining context-dependent candidates, LLM-guided falsification produces targeted non-reasoning inputs that trigger activation and meaning-preserving paraphrases of top-activating reasoning traces that suppress it. A small steering study yields minimal changes on the evaluated benchmarks. Overall, our results suggest that, in the settings we study, sparse decompositions can favor low-dimensional correlates that co-occur with reasoning, underscoring the need for falsification when attributing high-level behaviors to individual SAE features. Code is available at https://github.com/GeorgeMLP/reasoning-probing.

[1]UC Berkeley [2]UCSF. Correspondence to: George Ma <george_ma@berkeley.edu>.

*Proceedings of the 43rd International Conference on Machine Learning*, Seoul, South Korea. PMLR 306, 2026. Copyright 2026 by the author(s).

## 1. Introduction

Recent advances in large language models (LLMs) have demonstrated strong performance on tasks that require multi-step reasoning (Huang et al., 2024; Ahn et al., 2024). These capabilities are often enabled by chain-of-thought (CoT) prompting and other inference-time strategies, which encourage LLMs to generate intermediate reasoning traces prior to producing a final answer (Wei et al., 2022; Yao et al., 2023). Motivated by these advances, a growing body of work has explored mechanistic interpretability approaches to identify and interpret the internal representations underlying reasoning in LLMs (Chen et al., 2025; Li et al., 2025b).

A prominent interpretability approach in this area is the use of sparse autoencoders (SAEs). SAEs learn sparse decompositions that disentangle polysemantic neuron activations into more monosemantic and human-interpretable features (Bricken et al., 2023; Templeton et al., 2024). When studying reasoning with SAEs, many prior works rely on contrastive methods to identify reasoning features. In this paradigm, SAE features are evaluated on contrastive datasets constructed from reasoning and non-reasoning text (e.g., CoT responses versus direct answers), and features exhibiting large activation differences are interpreted as reasoning features that encode reasoning processes (Chen et al., 2025; Galichin et al., 2025; Venhoff et al., 2025; Li et al., 2025b).

Despite their widespread use, these contrastive activation-based methods suffer from fundamental ambiguities. Because CoT responses differ systematically from direct answers not only in their underlying reasoning processes but also in surface lexical usage and stylistic patterns, features with large activation differences may reflect shallow lexical confounds, such as recurring discourse tokens (e.g., *Wait*, *But*, *Let*) rather than the model's internal reasoning computations. Therefore, activation differences alone are insufficient to determine whether an SAE feature captures genuine reasoning behavior.

To address this ambiguity, we present a theoretical analysis showing that sparsity in SAE decoding can induce an asymmetry between low- and high-dimensional structure. In a stylized setting where a high-level behavior admits many semantically equivalent realizations but co-occurs with a stable low-dimensional lexical cue, sparse decoding can preserve the cue coordinate while suppressing the remaining

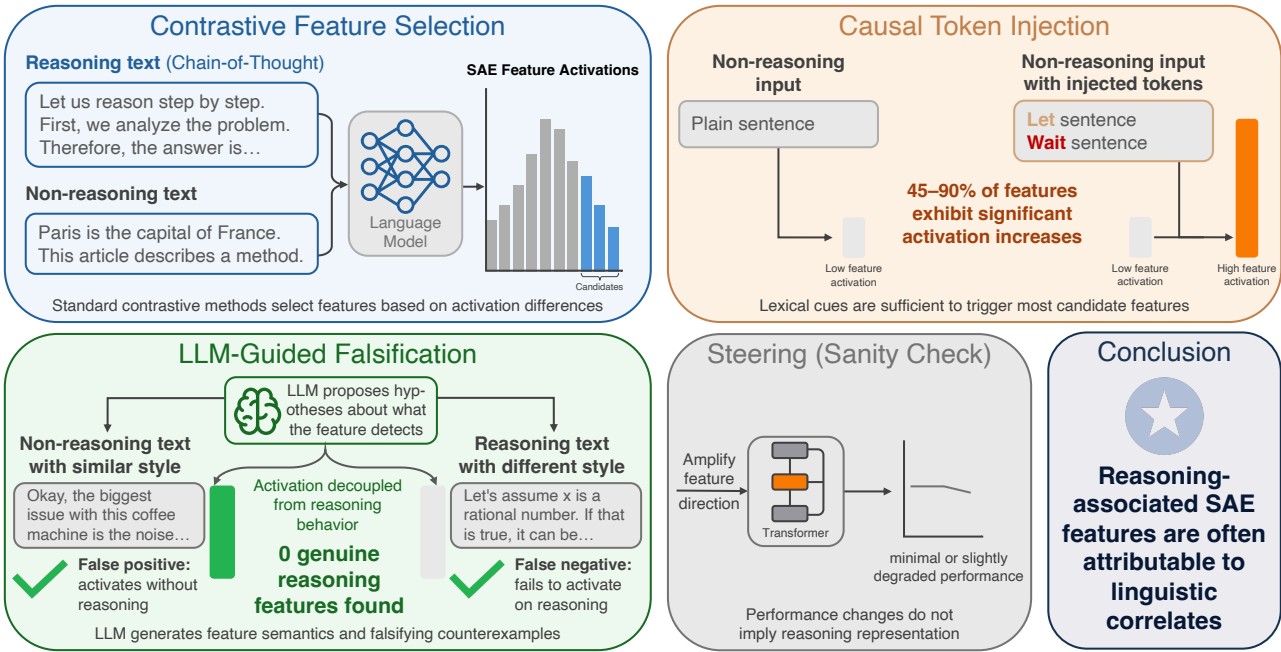

*Figure 1.* **Overview of our falsification-based evaluation framework for reasoning features.** We begin with SAE features identified by contrastive methods. We then test whether lexical cues are sufficient to induce activation via causal token injection. Remaining features are subjected to LLM-guided adversarial counterexample generation, which decouples reasoning behavior from feature activation. Across all configurations, this process fails to identify any feature that satisfies our criteria for genuine reasoning behavior.

distributed variation. Notably, we do not treat lexical patterns and reasoning as opposing: they are often coupled, and sparsity can cause SAE features to preferentially capture the low-dimensional part of that coupled signal.

Motivated by this mechanism, we introduce a falsification-based evaluation framework that probes whether contrastively selected "reasoning features" reflect cue-like structure or remain robust under increasingly targeted tests. We start from contrastive candidates and perform causal token injection by inserting a few feature-associated tokens into non-reasoning text to test whether lexical cues alone are sufficient to elicit activation. We then apply LLM-guided falsification to construct counterexamples: non-reasoning false positives (FP) that instantiate hypothesized lexical, syntactic, or discourse patterns, and false negatives (FN) obtained by paraphrasing a feature's top-activating reasoning traces to preserve meaning while suppressing activation. We also include a small steering study as a supplementary behavioral check. Figure 1 summarizes the pipeline.

Across 22 configurations, token injection shows that many contrastive candidates are elicited by inserting only a few associated tokens or short contexts into non-reasoning text. For the remaining features, LLM-guided falsification reliably constructs targeted FPs and meaning-preserving FNs that separate activation from the presence of a reasoning trace in the input. Under our operational criteria, we do not observe features that robustly track reasoning across these

probes. Together, these results suggest sparse decompositions can favor low-dimensional correlates that co-occur with reasoning, underscoring the need for falsification when attributing high-level behaviors to individual SAE features.

## 2. Related Work

### 2.1. Mechanistic Interpretability and SAEs

Mechanistic interpretability aims to reverse-engineer LLMs by analyzing and decomposing their internal representations (Sharkey et al., 2025). Building on the linear representation hypothesis (Elhage et al., 2022; Park et al., 2024), SAEs have emerged as a prominent tool for disentangling polysemantic neurons into more monosemantic, human-interpretable features, by learning a sparse decomposition of model activations (Bricken et al., 2023; Cunningham et al., 2023; Templeton et al., 2024). Recent work has further explored a range of SAE variants and training strategies, leading to improved feature quality and interpretability (Chanin et al., 2025; Leask et al., 2025; Li et al., 2025a). Empirical studies report that SAE features can align with semantically coherent content patterns, including programming-related text (Bricken et al., 2023), arithmetic concepts (Engels et al., 2025), and safety-related content (O'Brien et al., 2025). Despite these findings, the interpretability of SAE features has primarily been shown through easily identifiable textual patterns, leaving open the question of whether these features

capture deeper, higher-level aspects of model behavior.

## 2.2. SAEs for Reasoning in LLMs

Recent LLMs, such as OpenAI o1 (Jaech et al., 2024) and DeepSeek-R1 (Guo et al., 2025), are trained to elicit multi-step reasoning, yielding substantial performance gains on complex tasks (Wang et al., 2023; Wei et al., 2022). Motivated by these advances, several recent works apply SAEs to identify internal features associated with reasoning processes. One line of work applies contrastive activation-based methods to identify features that exhibit large activation differences or correlate with the presence of reasoning text. Steering such features has been shown to increase reasoning length and model confidence (Venhoff et al., 2025; Chen et al., 2025; Li et al., 2025b). Complementary approaches define sets of reasoning-related vocabularies and then identify reasoning features either by selective activation on these tokens (Galichin et al., 2025) or by exhibiting strong positive logit contributions to them (Fang et al., 2026). However, existing approaches implicitly assume that features correlated with reasoning-style text reflect underlying reasoning processes. In practice, it remains unclear whether such correlations reflect the model's internal reasoning computations or instead arise from superficial token-level confounds.

## 3. Theory: Sparsity Biases SAEs Toward Low-Dimensional Correlates

SAEs minimize reconstruction error under an explicit sparsity pressure, commonly implemented as an $\ell_1$ penalty on feature activations (Tibshirani, 1996; Olshausen & Field, 1997). In this section, we use a minimal model to motivate why this objective can preferentially surface token-level correlates of reasoning traces, even when the underlying reasoning behavior is present and correlated with those cues. This motivates the falsification pipeline developed later.

We model reasoning activations as having two parts that co-occur in typical CoT data. The first is a stable low-dimensional cue that reliably appears in reasoning traces, such as a recurring token like *wait*. The second is a high-dimensional component that varies across semantically equivalent realizations of a reasoning trace, capturing differences in phrasing, decomposition into steps, and local rhetorical structure. Formally, let $\{v_1, \ldots, v_k\} \subset \mathbb{R}^d$ be an orthonormal set and define a reasoning activation vector

$$h := a v_1 + b g, \quad g \in \mathrm{span}\{v_2, \ldots, v_k\} \quad (1)$$

where $a \geq 0$ and $b > 0$ are scalars and $g \sim \mathcal{N}\left(0, \frac{\mathbf{I}_{k-1}}{k-1}\right)$. We interpret $v_1$ as a cue direction shared across many reasoning traces, while the remaining component is distributed across the other $k - 1$ directions. The Gaussian model for $g$ serves as a tractable proxy for high-dimensional within-reasoning variability.

To isolate the effect of sparsity, we analyze the decoding step induced by an $\ell_1$ objective. Concretely, we consider the best-case setting where the decoder columns span the true reasoning subspace, so that without any sparsity pressure the least-squares code would recover $h$ exactly (zero reconstruction error) by projecting onto this basis. This lets us attribute any loss of the high-dimensional component to the sparsity objective rather than dictionary mismatch. Let $\mathbf{W} := [v_1, \ldots, v_k] \in \mathbb{R}^{d \times k}$ and consider the $\ell_1$-regularized code

$$z^\star(h) \in \arg\min_{z \in \mathbb{R}^k} \|h - \mathbf{W} z\|_2^2 + \lambda \|z\|_1. \quad (2)$$

The following theorem shows that, for fixed sparsity level $\lambda/b$, the recovered energy from the high-dimensional component $b g$ is exponentially suppressed as the intrinsic dimension $k - 1$ grows. In contrast, Appendix B.2 shows that the $v_1$ coordinate is comparatively easy to retain.

**Theorem 3.1** (High-dimensional residual is suppressed by $\ell_1$ decoding). *Fix integers $k \geq 2$ and $d \geq k$. Let $\mathbf{W} = [v_1, \ldots, v_k]$ have orthonormal columns. Let $h$ be generated by (1), and $z^\star(h)$ be any minimizer of (2). Define $z^\star_{2:k} := (z^\star_2, \ldots, z^\star_k) \in \mathbb{R}^{k-1}$. Then with $u := \frac{\lambda \sqrt{k-1}}{2b}$, we have*

$$\mathbb{E}\left[\|z^\star_{2:k}\|_2^2\right] \leq b^2 \Psi(u), \qquad \Psi(u) := 2\phi(u)/u,$$

*where $\phi$ is the standard normal density. In particular,*

$$\Psi(u) = \sqrt{2/\pi} \exp\left(-u^2/2\right)/u,$$

*so the expected recovered energy from the $(k - 1)$-dimensional component decays exponentially in $(k - 1)\lambda^2$.*

Theorem 3.1 captures the core asymmetry underlying our empirical pipeline. When a high-level behavior co-occurs with a stable cue, sparse decoding can explain a large portion of the contrastive signal using the low-dimensional coordinate alone, while distributing the remaining behavior-specific variation across many small coordinates that are individually penalized. This provides a concrete mechanism by which contrastive feature detection can identify features that separate reasoning from non-reasoning, yet still fail to isolate a single feature that tracks reasoning itself. Unlike feature absorption (Chanin et al., 2025) (nested-feature competition), our analysis isolates a dimensionality effect: sparsity penalizes representing high-dimensional within-behavior variability more than retaining a stable low-dimensional correlate, even under a perfect dictionary. Full proofs and an analogous theorem for Top-$K$ activations are provided in Appendix B.

Our model is intentionally simplified. It abstracts reasoning as high-dimensional variation within a subspace and analyzes an idealized sparse decoding objective rather than full SAE training dynamics. The value of the theory is therefore

motivational: it isolates a sparsity-driven failure mode that makes token-level correlates attractive to SAEs when they co-occur with richer behavior, which is precisely what our falsification experiments are designed to test.

## 4. Methodology

This section specifies (i) an operational definition of *genuine reasoning features* in SAEs, (ii) a contrastive method for identifying candidate reasoning features, and (iii) causal injection and LLM-based falsification experiments to distinguish genuine reasoning features from superficial correlates.

### 4.1. Defining Reasoning Features in SAEs

In this section, we define what we term a *genuine reasoning feature* in SAEs. Such a feature should reflect underlying reasoning processes, such as logical deduction or counterfactual reasoning, rather than surface-level correlates of reasoning-style text. Crucially, it is defined at the level of semantic computation rather than lexical form.

Formally, let $x \in \mathbb{R}^{d_{\text{model}}}$ denote a residual stream activation at layer $\ell$ of a transformer language model, and let $f = \text{SAE}_{\text{enc}}(x) \in \mathbb{R}^{d_{\text{SAE}}}$ be the output of an SAE encoder. We refer to $f_i$ as *feature $i$*. We say that $f_i$ is a *genuine reasoning feature* iff it satisfies all of the following criteria:

**Reasoning specificity.** The feature activates reliably on text that involves reasoning behavior, while exhibiting substantially lower activation on non-reasoning text.

**Non-spurious correlation.** The feature does not activate on non-reasoning text that merely contains phrases associated with reasoning corpora, including discourse markers such as "therefore" or "let us consider," as well as other surface-level lexical artifacts frequently found in CoT text.

**Semantic invariance.** The feature remains stable under paraphrases and stylistic transformations that preserve the underlying reasoning structure. That is, changing tone, vocabulary, or presentation style without altering the logical content should not substantially alter the feature's activation.

This definition is intentionally strict: it excludes features whose apparent "reasoning selectivity" is explained by shallow lexical or stylistic regularities, rather than by a stable representation of the underlying reasoning process. This distinction matters because SAE features can sharply separate reasoning from non-reasoning while encoding only a low-dimensional cue. For example, a feature that primarily detects a token such as "wait" can score highly under contrastive selection, and steering it can increase the model's tendency to emit that cue, potentially changing downstream generation style without implying that the feature represents reasoning computations such as reflection or backtracking.

We therefore treat contrastive correlations as hypotheses and use causal and adversarial tests to evaluate whether an interpretation survives meaning-preserving paraphrase and cue-matched counterexamples; see Appendix A for a detailed discussion.

### 4.2. Identifying Candidate Reasoning Features

Following prior work (Venhoff et al., 2025; Chen et al., 2025; Li et al., 2025b), we start by identifying candidate features that satisfy *Reasoning specificity* using contrastive activation-based methods applied to reasoning and non-reasoning text. We then consider a range of metrics to select top-ranked features as reasoning-related candidates.

Let $\mathcal{D}_{\text{R}}$ and $\mathcal{D}_{\text{NR}}$ denote corpora of reasoning and non-reasoning text, respectively. For each input sequence $x$, we aggregate each feature's activation across token positions by taking the maximum: $a_i(x) = \max_t f_i(x, t)$, where $f_i(x, t)$ denotes the activation of feature $i$ at token position $t$ for input $x$. We use a maximum aggregator to retain the single most salient activation while reducing sensitivity to sequence length and repeated occurrences of the same cue. Applying this aggregation to all samples yields two empirical activation distributions, $\{a_{i,j}^{\text{R}}\}_{j=1}^{n_{\text{R}}}$ and $\{a_{i,k}^{\text{NR}}\}_{k=1}^{n_{\text{NR}}}$, corresponding to reasoning and non-reasoning corpora. Candidate reasoning features are those whose activation distributions differ substantially between these two sets.

Our primary metric for selecting candidate features is Cohen's $d$, which measures the standardized mean difference between two distributions (Cohen, 2013). Specifically, for each feature $i$, we compute

$$d_i = \frac{\bar{a}_i^{\text{R}} - \bar{a}_i^{\text{NR}}}{s_{\text{pooled}}},$$

where $s_{\text{pooled}} = \sqrt{\frac{(n_{\text{R}}-1)(s_i^{\text{R}})^2 + (n_{\text{NR}}-1)(s_i^{\text{NR}})^2}{n_{\text{R}}+n_{\text{NR}}-2}}$, and $\bar{a}_i^{\text{R}}$ and $\bar{a}_i^{\text{NR}}$ denote the sample means of $\{a_{i,j}^{\text{R}}\}_{j=1}^{n_{\text{R}}}$ and $\{a_{i,k}^{\text{NR}}\}_{k=1}^{n_{\text{NR}}}$, respectively, with $s_i^{\text{R}}$ and $s_i^{\text{NR}}$ denoting the corresponding sample standard deviations.

Cohen's $d$ provides a scale-invariant measure of effect size, reflecting the practical significance of activation differences. We rank all SAE features by $d_i$ and select the top $k$ features as candidate reasoning features, with $k = 100$ in all experiments. In addition to Cohen's $d$, we assess robustness using two additional metrics: (i) ROC-AUC when using $a_i(x)$ as a score to discriminate between reasoning and non-reasoning samples; and (ii) activation frequency ratio on reasoning and non-reasoning samples. We observe consistent results across metrics, and provide detailed definitions and analyses for these additional metrics in Appendix D.

## 4.3. Causal Token Injection Framework

In this section, we test whether the candidate features identified in Section 4.2 satisfy the *Non-spurious correlation* criterion through causal token injection experiments. Our central hypothesis is straightforward: if a feature encodes genuine reasoning, then inserting a small number of its most activating tokens into non-reasoning text should not substantially increase its activation.

For each candidate feature $f_i$, we start by identifying the tokens and short token sequences that most strongly activate it on the reasoning corpus.

**Unigram ranking.** For each unique token $t$, we compute its mean activation

$$\bar{f}_{i,t} = \frac{1}{|\mathcal{I}_t|} \sum_{(\boldsymbol{x},j) \in \mathcal{I}_t} f_i(\boldsymbol{x},j),$$

where $\mathcal{I}_t = \{(\boldsymbol{x},j) \colon \text{token}_{\boldsymbol{x},j} = t\}$ denotes the set of all token positions at which token $t$ occurs in the reasoning corpus. Tokens are then ranked by $\bar{f}_{i,t}$ in descending order.

**Bigram and trigram ranking.** We extend the same procedure to contiguous bigrams and trigrams. We restrict attention to bigrams that appear at least three times and trigrams at least two times. Mean activations are computed analogously and ranked within each $n$-gram category.

We then evaluate whether the lexical patterns identified above are sufficient to drive feature activation outside reasoning contexts. For each feature $f_i$, we construct two sets of samples: *Baseline*, consisting of non-reasoning inputs drawn from $\mathcal{D}_{\text{NR}}$, and *Injected*, consisting of the same non-reasoning inputs with top-activating tokens inserted.

**Injection strategies.** To probe different forms of lexical sensitivity, we consider various injection strategies. First, *simple token injection* inserts three top-ranked tokens either by prepending them to the text, uniformly interspersing them throughout the text, or replacing randomly selected words. Second, *n-gram injection* inserts either two top-ranked bigrams or one top-ranked trigram at uniformly sampled positions within the text. Third, *contextual injection* inserts tokens along with their frequent preceding context to preserve local co-occurrence structure.

**Feature Classification.** For each feature $i$, we quantify the activation shift induced by injection using Cohen's $d$:

$$d_i^{\text{injected}} = \frac{\bar{a}_i^{\text{injected}} - \bar{a}_i^{\text{baseline}}}{s_{\text{pooled}}},$$

where $\bar{a}_i^{\text{injected}}$ and $\bar{a}_i^{\text{baseline}}$ denote the mean activations on injected and baseline samples, respectively, and $s_{\text{pooled}}$ is the pooled standard deviation.

We assess statistical significance using independent two-sample $t$-tests comparing injected and baseline activations. Following standard conventions for effect size interpretation (Cohen, 2013), each feature is classified based on the strongest effect observed across all injection strategies:

1. *Token-driven*: $d \geq 0.8$ with $p < 0.01$, corresponding to a large effect.
2. *Partially token-driven*: $0.5 \leq d < 0.8$ with $p < 0.01$, corresponding to a medium effect.
3. *Weakly token-driven*: $0.2 \leq d < 0.5$ with $p < 0.05$, corresponding to a small effect.
4. *Context-dependent*: $d < 0.2$ or $p \geq 0.05$, indicating a negligible effect.

## 4.4. LLM-Guided Falsification

The token injection experiments described above eliminate a large class of lexical and short-range contextual confounds. Features classified as context dependent, however, are not well explained by token-level triggers alone and may instead respond to higher-level semantic patterns that are difficult to probe through token manipulations. We therefore apply LLM-guided falsification to test whether these features satisfy the *Semantic invariance* criterion by constructing targeted counterexamples that attempt to falsify proposed explanations. These counterexamples include non-reasoning text designed to induce spurious activation, as well as paraphrases of previously activating reasoning text that fail to elicit activation. We detail this process next.

**Hypothesis generation.** The LLM is first provided with three sources of information: the top-activating tokens identified in Section 4.3, a small set of reasoning samples that strongly activate the feature, and token-level activation traces within those samples. Using this information, the LLM proposes an explicit hypothesis describing what linguistic or structural pattern the feature may be detecting.

**Counterexample construction.** Given a proposed hypothesis, the LLM generates two complementary types of test cases. First, it produces non-reasoning text that instantiates the hypothesized pattern, serving as candidate false positives (FP). Second, for false negatives (FN), it paraphrases the feature's top-activating reasoning samples while explicitly preserving their semantic content and reasoning intent, but altering surface form such as vocabulary, syntax, or presentation style. Each candidate is evaluated by running it through the target model and measuring the maximum activation of the feature. A non-reasoning candidate is considered a valid FP if its activation exceeds a threshold $\tau = 0.5 \times \max_j a_i(\boldsymbol{x}_j^{\text{R}})$, while a paraphrased reasoning candidate is considered a valid FN if its activation remains below $0.1\tau$, where the maximum is taken over the original reasoning samples. The generation steps are repeated for up

to $T = 10$ iterations, with early termination if at least three valid FP and three valid FN are obtained.

**Final classification.** Given the constructed counterexamples, the LLM produces a consolidated interpretation of the feature, including the linguistic or structural pattern it appears to detect and the conditions under which it activates or fails to activate. A final classification is then made using all evidence accumulated during the procedure, including the success or failure of generated counterexamples, activation measurements on those candidates, and consistency across iterations. To ensure reliability, we report all LLM-generated interpretations and counterexamples for a representative experiment in Appendix G.

### 4.5. Steering Experiments

As a supplementary analysis, we conduct steering experiment to probe whether amplifying candidate features identified in Section 4.2 has a measurable effect on downstream reasoning task performance.

**Feature Steering.** For a feature $f_i$ with decoder direction $\mathbf{W}_{\text{dec},i} \in \mathbb{R}^{d_{\text{model}}}$, we intervene on the residual stream at layer $\ell$ according to $\boldsymbol{x}' = \boldsymbol{x} + \gamma f_i^{\max} \mathbf{W}_{\text{dec},i}$, where $\gamma \in \mathbb{R}$ is the steering strength and $f_i^{\max}$ is the maximum activation of $f_i$ on the reasoning corpus.

## 5. Experiments

### 5.1. Experimental Setup

We conduct experiments on six open-weight transformer language models. Our primary models are Gemma-3-12B-Instruct and Gemma-3-4B-Instruct (Gemma Team et al., 2025). For both, we analyze a subset of middle-to-late layers and use SAEs from the GemmaScope-2 release (McDougall et al., 2025), each with 16,384 features trained on residual stream activations. We additionally study DeepSeek-R1-Distill-Llama-8B (Guo et al., 2025), analyzing a representative middle layer using an SAE trained on reasoning-oriented corpora, specifically LMSys-Chat-1M (Zheng et al., 2024) and OpenThoughts-114k (Guha et al., 2025), with 65,536 features (Galichin et al., 2025). Results for Llama-3.1-8B (Grattafiori et al., 2024), Gemma-2-2B and Gemma-2-9B (Gemma Team et al., 2024) are reported in Appendix C.

For reasoning corpora, we use s1K-1.1, which consists of 1,000 challenging mathematics problems with detailed CoT traces (Muennighoff et al., 2025), and the General Inquiry Thinking Chain-of-Thought dataset, which contains 6,000 question-answer pairs spanning diverse domains with explicit reasoning annotations (Wensey, 2025). As a non-reasoning corpus, we use an uncopyrighted subset of the

Pile (Gao et al., 2020), a large-scale collection of general web text. From each corpus, we uniformly sample 1,000 texts and chunk inputs to 64 tokens.

We focus on middle-to-late layers, motivated by both prior work and empirical analysis. Early layers are typically dominated by lexical processing, while late layers are increasingly specialized toward output token prediction (Ma et al., 2025). To further support this choice, we analyze how concentrated SAE feature activations are on individual tokens. Specifically, we compute a token concentration ratio, defined as the fraction of a feature's activation mass attributable to its 30 most activating tokens, and the normalized entropy of the activation distribution across tokens.

Figure 2 shows diagnostics for Gemma-3-4B-Instruct on the s1K dataset. Middle layers consistently exhibit lower concentration ratios and higher entropy than early or late layers. Similar trends are observed across models. If genuine reasoning exists at the level of individual features, these layers constitute the most plausible candidates.

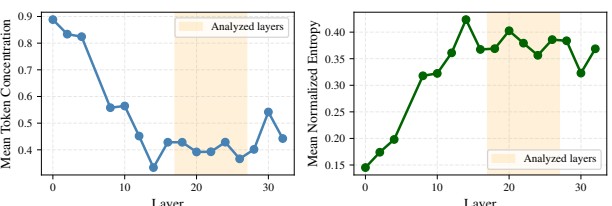

*Figure 2.* Token concentration ratio and normalized activation entropy for SAE features across all layers of Gemma-3-4B-Instruct on the s1K dataset. Middle layers exhibit lower concentration and higher entropy, indicating reduced reliance on specific tokens.

We conduct all experiments on a single NVIDIA A100 GPU with 80 GB VRAM. Feature detection and token injection experiments require around 1 hour per configuration, while LLM-guided falsification requires approximately 2 hours.

### 5.2. Feature Detection Results

We apply the contrastive procedure described in Section 4.2 to identify candidate reasoning features. For each configuration, we rank SAE features by Cohen's $d$ and select the top 100 features. Results obtained using alternative metrics, including ROC-AUC and activation frequency ratios, are reported in Appendix D and are qualitatively consistent.

Table 1 reports the mean Cohen's $d$ of the top 100 features for each configuration. Mean effect sizes range from 0.675 to 1.043, corresponding to medium to large effects under standard conventions (Cohen, 2013). This indicates that SAE features can reliably distinguish between reasoning and non-reasoning text at a statistical level.

Figure 3 shows the full distribution of Cohen's $d$ values for Gemma-3-12B-Instruct at layer 22 on the s1K dataset.

*Table 1.* Mean Cohen's $d$ values for the top 100 features.

| Model | Layer | s1K | General Inquiry CoT |
|---|---|---|---|
| Gemma-3-12B-Instruct | 17 | 0.775 | 0.947 |
| Gemma-3-12B-Instruct | 22 | 0.835 | 0.924 |
| Gemma-3-12B-Instruct | 27 | 0.805 | 0.995 |
| Gemma-3-4B-Instruct | 17 | 0.909 | 0.984 |
| Gemma-3-4B-Instruct | 22 | 0.892 | 1.043 |
| Gemma-3-4B-Instruct | 27 | 0.888 | 1.010 |
| DS-R1-Distill-Llama-8B | 19 | 0.675 | 0.781 |

The distribution exhibits a long tail, with a small subset of features achieving substantially larger effect sizes than the majority. These top-ranked features form a clearly separated region in the right tail, motivating their selection for further falsification analyses.

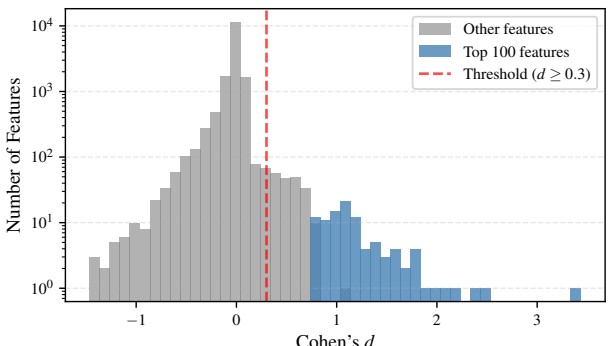

*Figure 3.* Distribution of Cohen's $d$ values across SAE features for Gemma-3-12B-Instruct at layer 22.

## 5.3. Token Injection Results

We apply the token injection procedure described in Section 4.3 to the top 100 candidate reasoning features per configuration. For each feature, we test whether inserting its most activating tokens into non-reasoning text is sufficient to elicit strong activation, and assign the feature to the strongest classification observed across all injection strategies, following Section 4.3.

Table 2 summarizes the resulting classifications. Across all models, layers, and datasets, the majority of features exhibit substantial activation increases under token injection alone, indicating strong sensitivity to token-level patterns.

Across configurations, between 45% and 90% of features are classified as token-driven, partially token-driven, or weakly token-driven, demonstrating statistically significant activation increases induced by token injection alone. Only a minority of features remain context-dependent after injection. As reported in Appendix E.3, mean Cohen's $d$ values range from 0.521 to 1.471 across configurations, indicating that inserting a small number of tokens into non-reasoning

text is sufficient to produce substantial activation changes.

Figure 4 visualizes these results across all configurations. The overall pattern is consistent across models and layers. Gemma-3-12B-Instruct evaluated on the s1K dataset exhibits the largest fraction of context-dependent features at 55%, while several Gemma configurations on the same dataset exhibit fewer than 12%.

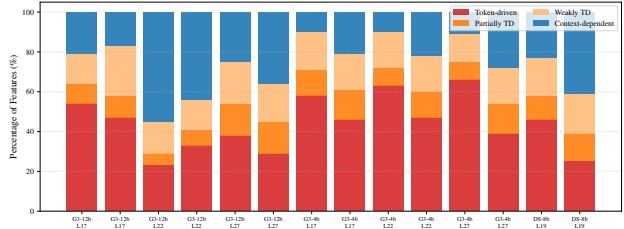

*Figure 4.* Distribution of token injection classifications across configurations. Each bar corresponds to a model-layer configuration, with segments indicating the proportion of features classified as token-driven (TD), partially TD, weakly TD, or context-dependent.

Additional statistics are reported in Appendix E. Overall, these results indicate that most candidate reasoning features are driven by superficial lexical patterns. We therefore subject the remaining context-dependent features to the LLM-guided falsification analysis described next.

## 5.4. LLM-Guided Falsification Results

We apply the LLM-guided falsification procedure described in Section 4.4 to features that remain context dependent after token injection. For configurations with more than 20 such features, we randomly sample 20 for analysis. All falsification experiments are conducted using Gemini 3 Pro as the LLM hypothesis generator and interpreter.

Table 3 and 4 summarize the results for the s1K and General Inquiry CoT datasets, respectively. For each configuration, we report the number of analyzed features, the number classified as genuine reasoning features, and the number classified as confounds with high confidence.

Across all 248 context-dependent features analyzed, none are classified as genuine reasoning features. Each feature admits systematic false positives and false negatives under the LLM-guided protocol, with most classified as confounds with high confidence due to repeated success in generating falsifying counterexamples. These outcomes are consistent with the interpretation that feature activations can be driven by correlated linguistic or discourse patterns rather than reliably tracking reasoning behavior.

Qualitative analysis reveals recurring categories of confounds. Many features respond to instructional or planning-oriented discourse openers, such as *"Let's break down"* or *"I need to figure out"*, while others are triggered by formal

*Table 2.* Token injection classification results for the top 100 features per configuration. TD denotes token-driven, PTD partially token-driven, WTD weakly token-driven, and CD context-dependent.

| Model | Layer | s1K | | | | General Inquiry CoT | | | |
|---|---|---|---|---|---|---|---|---|---|
| | | TD | PTD | WTD | CD | TD | PTD | WTD | CD |
| Gemma-3-12B-Instruct | 17 | 54 | 10 | 15 | 21 | 47 | 11 | 25 | 17 |
| Gemma-3-12B-Instruct | 22 | 23 | 6 | 16 | 55 | 33 | 8 | 15 | 44 |
| Gemma-3-12B-Instruct | 27 | 38 | 16 | 21 | 25 | 29 | 16 | 19 | 36 |
| Gemma-3-4B-Instruct | 17 | 58 | 13 | 19 | 10 | 46 | 15 | 18 | 21 |
| Gemma-3-4B-Instruct | 22 | 63 | 9 | 18 | 10 | 47 | 13 | 18 | 22 |
| Gemma-3-4B-Instruct | 27 | 66 | 9 | 14 | 11 | 39 | 15 | 18 | 28 |
| DS-R1-Distill-Llama-8B | 19 | 46 | 12 | 19 | 23 | 25 | 14 | 20 | 41 |

*Table 3.* LLM-guided falsification results for context-dependent features on the s1K dataset.

| Model | Layer | Analyzed | Genuine | High Conf. |
|---|---|---|---|---|
| Gemma-3-12B-Instruct | 17 | 20 | 0 | 20 |
| Gemma-3-12B-Instruct | 22 | 20 | 0 | 18 |
| Gemma-3-12B-Instruct | 27 | 20 | 0 | 20 |
| Gemma-3-4B-Instruct | 17 | 10 | 0 | 10 |
| Gemma-3-4B-Instruct | 22 | 10 | 0 | 10 |
| Gemma-3-4B-Instruct | 27 | 11 | 0 | 11 |
| DS-R1-Distill-Llama-8B | 19 | 20 | 0 | 14 |

*Table 4.* LLM-guided falsification results for context-dependent features on the General Inquiry CoT dataset.

| Model | Layer | Analyzed | Genuine | High Conf. |
|---|---|---|---|---|
| Gemma-3-12B-Instruct | 17 | 17 | 0 | 17 |
| Gemma-3-12B-Instruct | 22 | 20 | 0 | 19 |
| Gemma-3-12B-Instruct | 27 | 20 | 0 | 18 |
| Gemma-3-4B-Instruct | 17 | 20 | 0 | 20 |
| Gemma-3-4B-Instruct | 22 | 20 | 0 | 20 |
| Gemma-3-4B-Instruct | 27 | 20 | 0 | 20 |
| DS-R1-Distill-Llama-8B | 19 | 20 | 0 | 19 |

*Table 5.* Steering results on Gemma-3-12B-Instruct at layer 22 using the top three features on the s1K dataset.

| Feature | AIME Baseline | AIME Steered | GPQA Baseline | GPQA Steered |
|---|---|---|---|---|
| 1053 | 26.7% | 20.0% | 37.9% | 13.6% |
| 0 | 26.7% | 10.0% | 37.9% | 20.2% |
| 578 | 26.7% | 26.7% | 37.9% | 33.3% |

As discussed in Section 4.1, steering performance is not a reliable indicator of whether a feature encodes reasoning. Prior work shows that simple lexical interventions can substantially improve these benchmarks without engaging reasoning mechanisms (Muennighoff et al., 2025). Consistent with our broader results, steering yields minimal changes or degradations in accuracy.

## 6. Summary

In this work, we study whether SAEs identify genuine reasoning features in LLMs. Motivated by a stylized analysis suggesting that sparsity can favor stable low-dimensional correlates over high-dimensional within-reasoning variation, we develop a falsification-oriented evaluation framework that combines causal token injection with LLM-guided counterexample generation. Across 22 configurations, most contrastively selected candidates are highly sensitive to injecting only a few associated tokens, and the remaining features admit targeted counterexamples that decouple activation from the presence of a reasoning trace. Overall, our results suggest that contrastive correlations are not, by themselves, sufficient evidence for monosemantic reasoning features, and that causal validation is important when attributing high-level behaviors to individual SAE directions.

## 7. Limitations

Our conclusions are scoped to the contrastive candidate features and experimental settings we study, and primarily address monosemantic interpretations of individual SAE features selected by activation-based criteria. They do not rule out non-monosemantic features that mix cue-like patterns with aspects of reasoning, nor the possibility that reasoning-

academic framing common in explanatory writing, including phrases like *"We are given"* or *"The problem asks"*. In each case, the feature activates on non-reasoning text that matches the pattern and fails to activate on reasoning text that avoids it. All interpretations and counterexamples for a representative experiment are reported in Appendix G.

Overall, even features that are not explained by simple token injection can still admit targeted counterexamples under falsification, suggesting that caution is warranted when attributing such features to reasoning itself.

### 5.5. Steering Results

We report a small-scale steering experiment as a supplementary analysis, following the setup in Section 4.5. We focus on Gemma-3-12B-Instruct at layer 22 and select the top three features on the s1K dataset. For each feature $f_i$, we apply steering to the residual stream with $\gamma \in \{0, 2\}$. Table 5 reports one-shot accuracy under these conditions.

relevant information is distributed across many features or represented nonlinearly in ways that resist single-feature attribution. Accordingly, we view our results as guidance on what current contrastive pipelines reliably establish, and we recommend treating contrastive correlations as hypotheses that warrant causal and adversarial validation.

## Acknowledgement

This research was supported by the U.S. Army Research Laboratory and the U.S. Army Research Office under Grant W911NF2010219, the Office of Naval Research, the National Science Foundation, Google, and Apple Machine Learning. This work used Jetstream2 (Hancock et al., 2021; Boerner et al., 2023) at Indiana University through allocation CIS240843 from the Advanced Cyberinfrastructure Coordination Ecosystem: Services & Support (ACCESS) program, which is supported by National Science Foundation grants #2138259, #2138286, #2138307, #2137603, and #2138296.

## Impact Statement

This paper presents work whose goal is to advance the field of Machine Learning. There are many potential societal consequences of our work, none which we feel must be specifically highlighted here.

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

# Appendix

## Table of Contents

## A. On interpreting high-level behaviors with sparse autoencoders

A growing body of work uses sparse autoencoders (SAEs) to study reasoning in large language models by identifying features whose activations differ between reasoning and non-reasoning corpora and then interpreting these directions as reasoning-related features (Galichin et al., 2025; Venhoff et al., 2025; Chen et al., 2025; Li et al., 2025b; Wang et al., 2025; Fang et al., 2026; Zhang et al., 2025; Dong et al., 2026; He et al., 2026; Li et al., 2026). These results provide strong evidence that SAEs surface directions that reliably separate reasoning-style text from other generations, and they have enabled detailed case studies and interventions. Our results refine how to read such separations. In the main text, we present a stylized theoretical analysis showing that sparsity regularization intrinsically favors stable low-dimensional correlates over high-dimensional within-behavior variation, even when the dictionary contains the relevant directions. When a high-level behavior is tightly coupled with a recurring cue, sparse coding can retain the cue cheaply while suppressing the remaining distributed component. This mechanism motivates why contrastive separability can coexist with a lack of monosemantic features that track the underlying reasoning process. Figure 5 illustrates the core intuition: an SAE feature that detects a cue token can separate reasoning from non-reasoning and can be steered to increase the probability of emitting that cue, yet still encode no semantic information about the reasoning itself. Consistent with this picture, our falsification experiments go beyond distributional contrasts by combining causal token injection with LLM-guided counterexamples, constructing both *false positives* that elicit activation in non-reasoning text via associated token-level patterns and *false negatives* obtained by paraphrasing top-activating reasoning samples to preserve meaning while suppressing activation. Together, these tests provide direct evidence that many contrastively selected features are better explained as linguistic correlates than as isolated representations of reasoning computations. This perspective also aligns with observations from test-time scaling, where substantial gains on reasoning benchmarks can arise from simple inference-time manipulations that change token usage and generation length (Muennighoff et al., 2025).

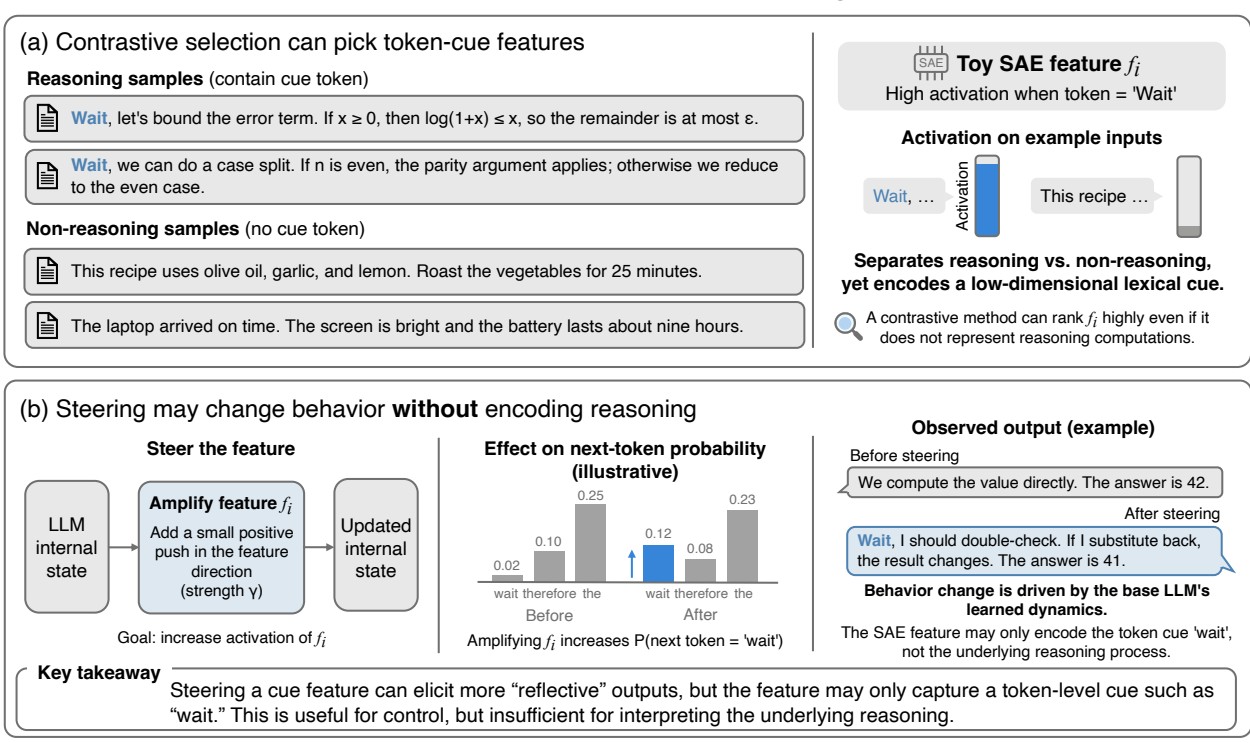

*Figure 5.* Illustration of cue-driven separability without semantic reasoning representation. **Top:** Reasoning traces often contain stable cue tokens such as *wait*, while non-reasoning text does not. A contrastively selected SAE feature that activates on the cue can separate reasoning from non-reasoning despite encoding only a low-dimensional lexical pattern. **Bottom:** Steering such a feature can increase the probability that the LLM emits the cue token, which may correlate with more reflective or backtracking-like outputs. This behavioral change reflects dynamics already present in the base LLM, and does not imply that the SAE feature represents the underlying reasoning process.

**Our framing is not that linguistic patterns and high-level behaviors are separate or opposing.** Reasoning traces inevitably involve language, and token-level structure can be a genuine part of how reasoning is expressed. The issue is coupling. When a recurring cue is strongly correlated with a broader behavioral state, sparsity creates an asymmetry in what is easiest to represent: low-dimensional correlates are retained by activating a small number of coordinates, while high-dimensional variability across semantically equivalent realizations is penalized because it requires distributing mass across many coordinates. This is conceptually different from feature absorption (Chanin et al., 2025). Feature absorption studies how, under sparsity, a feature that is a subset of another can dominate because it achieves lower sparsity cost (for example, a "short" feature outcompeting a broader "starts with s" feature). Our theoretical analysis instead highlights a dimensionality effect: sparsity penalizes representing an isotropic or otherwise high-dimensional component more heavily than a stable low-dimensional correlate, even when both are part of the same behavioral phenomenon and even under an idealized dictionary.

Related concerns and opportunities appear in SAE analyses of other high-level behaviors such as refusal, hallucination, and instruction following. For refusals, SAE features can predict and influence refusal behavior, while also correlating with specific refusal phrases (Yeo et al., 2025). For hallucination and knowledge awareness, latent directions associated with entity familiarity can causally affect whether a model abstains or hallucinates attributes (Ferrando et al., 2025). For instruction-following settings, SAEs can be used to steer performance on structured tasks (He et al., 2025). These works demonstrate the practical utility of SAE features as control levers and as diagnostic signals. Our results suggest a complementary methodological caution: when an SAE feature correlates with a high-level behavior, that correlation can arise because the feature captures a low-dimensional linguistic correlate of the behavior, rather than a monosemantic representation of the underlying computation. This does not reduce the value of the features for intervention or analysis, but it affects what kinds of semantic claims are warranted without falsification.

Our falsification results have important limitations. First, they primarily address monosemantic interpretations of individual contrastive SAE features. They do not rule out the existence of non-monosemantic features that jointly respond to both a token-level cue and aspects of reasoning behavior, nor do they exclude the possibility that reasoning-relevant information is spread across many features in a way that resists single-feature attribution. Second, while our theoretical analysis explains why sparse objectives can suppress high-dimensional within-reasoning variation, it is not a complete characterization of SAE training dynamics or of how distributed computations manifest across layers and time steps. As a result, our evidence should be read as guidance about what current contrastive pipelines reliably establish, not as a claim that SAEs cannot represent any reasoning-relevant structure. In practice, we advocate treating contrastive correlations as hypotheses and using causal and adversarial tests, including paraphrase-based falsification of top-activating samples, to assess whether an interpretation is robust.

Overall, our findings suggest a constructive direction for the literature that interprets SAE features for high-level behaviors. The goal is not to dismiss SAE-based interpretability, but to strengthen it. SAEs remain a powerful tool for surfacing consistent internal directions and for enabling targeted interventions (Galichin et al., 2025; Venhoff et al., 2025; Chen et al., 2025; Li et al., 2025b; Wang et al., 2025; Fang et al., 2026; Zhang et al., 2025; Dong et al., 2026; He et al., 2026; Li et al., 2026; Yeo et al., 2025; Ferrando et al., 2025; He et al., 2025). Our contribution is to clarify a sparsity-driven failure mode that is especially relevant when high-level behaviors co-occur with low-dimensional linguistic correlates, and to provide falsification-oriented evaluations that can help practitioners distinguish robust behavioral representations from cue-like correlates.

# B. Theory: Sparsity suppresses high-dimensional within-reasoning variation

## B.1. Setting and notation

Fix integers $k \geq 2$ and $d \geq k$. Let $\{v_1, \ldots, v_k\} \subset \mathbb{R}^d$ be an orthonormal set and define the dictionary matrix $\mathbf{W} :=$ $[v_1, \ldots, v_k] \in \mathbb{R}^{d \times k}$ so that $\mathbf{W}^\top \mathbf{W} = \mathbf{I}_k$. We consider a stylized "reasoning subspace" in which reasoning-related activations lie in $\mathrm{span}\{v_1, \ldots, v_k\}$, with $v_1$ capturing a stable low-dimensional correlate and the remaining directions capturing within-reasoning variability.

Let $a \geq 0$ and $b > 0$ be scalars. We define a random activation vector

$$h := av_1 + bg, \tag{3}$$

where $g \in \mathrm{span}\{v_2, \ldots, v_k\}$ is distributed as

$$g \sim \mathcal{N}\left(0, \frac{1}{k-1}\mathbf{I}_{k-1}\right) \tag{4}$$

in the coordinate system of the basis $\{v_2, \ldots, v_k\}$. Explicitly, let $\xi \in \mathbb{R}^{k-1}$ have i.i.d. $\mathcal{N}(0, \frac{1}{k-1})$ coordinates and set $g := \sum_{i=2}^{k} \xi_{i-1} v_i$. Then $h$ has coordinates

$$\mathbf{W}^\top h = x \in \mathbb{R}^k, \qquad x_1 = a, \qquad x_i = b\xi_{i-1} \text{ for } i \in \{2, \ldots, k\}. \tag{5}$$

In particular, for $i \in \{2, \ldots, k\}$,

$$x_i \sim \mathcal{N}\left(0, \sigma^2\right), \qquad \sigma^2 := \frac{b^2}{k-1},$$

and the $x_2, \ldots, x_k$ are independent.

We analyze two activation rules used in sparse autoencoders.

**(i) $\ell_1$-regularized decoding.** Given $\lambda > 0$, define the $\ell_1$-regularized code

$$z^\star(h) \in \underset{z \in \mathbb{R}^k}{\arg\min} \|h - \mathbf{W}z\|_2^2 + \lambda\|z\|_1. \tag{6}$$

**(ii) Top-$K$ decoding.** Given an integer $K \in \{0, 1, \ldots, k\}$, define the Top-$K$ approximation by keeping the $K$ largest-magnitude coordinates of $x = \mathbf{W}^\top h$ and zeroing the rest. Formally, let $S_K(x) \subseteq \{1, \ldots, k\}$ be any size-$K$ index set achieving the $K$ largest values of $|x_i|$. Define $z^{(K)} \in \mathbb{R}^k$ by

$$z_i^{(K)} := x_i \cdot \mathbb{1}[i \in S_K(x)]. \tag{7}$$

Then $\mathbf{W}z^{(K)}$ is the orthogonal projection of $h$ onto the span of the selected basis directions.

## B.2. Main theorems

We first record how the cue coordinate behaves under the same sparse decoding rules we analyze for the residual. This makes the setting in Equations (3) and (4) easier to interpret: the cue component $av_1$ is a single stable direction, while the behavioral component $bg$ is isotropic within a $(k-1)$-dimensional subspace. We then state and prove the $\ell_1$ suppression result used in the main text, followed by an analogous bound for Top-$K$.

**Lemma B.1** ($\ell_1$ recovers the cue by soft-thresholding). *Fix $k \geq 2$ and let $\mathbf{W} \in \mathbb{R}^{d \times k}$ have orthonormal columns with $\mathbf{W} = [v_1, \ldots, v_k]$. Let $h$ follow Equations (3) and (4) with parameters $a \geq 0$ and $b > 0$, and let $z^\star(h)$ be any minimizer of Equation (6). Then*

$$z_1^\star(h) = \mathrm{ST}(a, \lambda/2) = \max\left(a - \lambda/2, 0\right). \tag{8}$$

*In particular, if $a \geq \lambda/2$ then the cue reconstruction error along $v_1$ equals $(a - z_1^\star(h))^2 = (\lambda/2)^2$.*

**Lemma B.2** (Top-$K$ selects the cue when it exceeds the residual order statistic). *Fix $k \geq 2$ and let $\mathbf{W} \in \mathbb{R}^{d \times k}$ have orthonormal columns with $\mathbf{W} = [v_1, \ldots, v_k]$. Let $h$ follow Equations (3) and (4) with parameters $a \geq 0$ and $b > 0$, and let*

$z^{(K)}$ be the Top-$K$ code defined in Equation (7) with $K \geq 1$. Let $x_i := \langle v_i, h \rangle$ for $i \geq 2$, and let $X^2_{(1)} \geq \cdots \geq X^2_{(k-1)}$ denote the order statistics of $\{x^2_2, \ldots, x^2_k\}$. Then

$$z_1^{(K)}(h) = a \quad \text{whenever} \quad a^2 \geq X^2_{(K)}. \tag{9}$$

Moreover, with $\sigma^2 := \frac{b^2}{k-1}$,

$$\mathbb{P}\left(z_1^{(K)}(h) \neq a\right) \leq 2(k-1)\exp\left(-\frac{a^2}{2\sigma^2}\right). \tag{10}$$

Lemmas B.1 and B.2 formalize a basic asymmetry in our setting. The cue component is a single coordinate in the orthonormal dictionary and is therefore inexpensive to represent: under $\ell_1$, it is recovered up to the usual soft-threshold bias, and under Top-$K$, it is selected whenever its magnitude is not dominated by the $K$-th largest residual coordinate. In contrast, the residual component is spread isotropically across $k - 1$ coordinates. The theorems below quantify how sparsity suppresses the recovered energy from this high-dimensional residual, even though the cue is still recovered.

**Theorem B.3** ($\ell_1$ suppresses high-dimensional residual). *Fix $k \geq 2$ and let $W$ have orthonormal columns. Let $h$ follow Equations (3) and (4) with parameters $a \geq 0$ and $b > 0$, and let $z^\star(h)$ be any minimizer of Equation (6). Define $z^\star_{2:k} := (z^\star_2, \ldots, z^\star_k) \in \mathbb{R}^{k-1}$. Let $\sigma^2 = \frac{b^2}{k-1}$ and define $u := \frac{\lambda}{2\sigma} = \frac{\lambda\sqrt{k-1}}{2b}$. Then*

$$\mathbb{E}\left[\|z^\star_{2:k}\|_2^2\right] = (k-1)\mathbb{E}\left[\text{ST}(X, \lambda/2)^2\right] \leq b^2 \Psi(u),$$

*where $X \sim \mathcal{N}(0, \sigma^2)$, $\text{ST}(x, t) := \text{sign}(x)\max\{|x| - t, 0\}$, and*

$$\Psi(u) := \frac{2\phi(u)}{u}. \tag{11}$$

*In particular,*

$$\mathbb{E}\left[\|z^\star_{2:k}\|_2^2\right] \leq b^2\sqrt{\frac{2}{\pi}}\frac{1}{u}\exp\left(-\frac{u^2}{2}\right), \tag{12}$$

*which decays exponentially in $(k-1)\lambda^2/b^2$ for fixed $\lambda/b$.*

**Theorem B.4** (Top-$K$ captures at most a $K\log k/k$ fraction of isotropic residual). *Fix $k \geq 2$ and let $W \in \mathbb{R}^{d \times k}$ have orthonormal columns. Let $h$ follow Equations (3) and (4) with parameters $a \geq 0$ and $b > 0$, and let $z^{(K)}$ be the Top-$K$ code defined in Equation (7). Define the residual energy captured outside the cue coordinate as*

$$R_K := \left\|\left(z_2^{(K)}, \ldots, z_k^{(K)}\right)\right\|_2^2.$$

*Let $\sigma^2 := \frac{b^2}{k-1}$ and let $X_1, \ldots, X_{k-1}$ be i.i.d. $\mathcal{N}(0, \sigma^2)$. Then for any $K \in \{0, 1, \ldots, k-1\}$,*

$$\mathbb{E}[R_K] \leq \sigma^2 K\left(2\log\left(2(k-1)\right) + 2\right). \tag{13}$$

*Equivalently,*

$$\frac{\mathbb{E}[R_K]}{b^2} \leq \frac{K}{k-1}\left(2\log\left(2(k-1)\right) + 2\right). \tag{14}$$

Theorem B.4 implies that if $K$ is held fixed while $k$ grows, the expected recovered fraction from the isotropic residual vanishes. More generally, the recovered fraction is at most on the order of $K\log k/k$, so capturing a constant fraction of a high-dimensional isotropic component requires $K$ to scale nearly linearly in $k$ up to logarithmic factors. This formalizes the intuition that representing isotropic high-dimensional variation requires activating many coordinates, which sparse Top-$K$ codes cannot do when $K \ll k$.

### B.3. Proofs

**Proof of Lemma B.1.** Let $x := W^\top h \in \mathbb{R}^k$. Since $W$ has orthonormal columns,

$$\|h - Wz\|_2^2 = \|x - z\|_2^2 + \left\|\left(I - WW^\top\right)h\right\|_2^2.$$

The second term does not depend on $\boldsymbol{z}$, so minimizing Equation (6) is equivalent to minimizing $\|\boldsymbol{x} - \boldsymbol{z}\|_2^2 + \lambda\|\boldsymbol{z}\|_1$ over $\boldsymbol{z} \in \mathbb{R}^k$. This objective is separable across coordinates. Because $\boldsymbol{g} \in \mathrm{span}\{\boldsymbol{v}_2, \ldots, \boldsymbol{v}_k\}$, we have $x_1 = \langle \boldsymbol{v}_1, \boldsymbol{h} \rangle = a$. Therefore $z_1^\star$ is any minimizer of the scalar problem

$$\min_{z \in \mathbb{R}} (z - a)^2 + \lambda|z|.$$

The unique minimizer is the soft-threshold operator $z = \mathrm{ST}(a, \lambda/2)$. Since $a \geq 0$, this equals $\max(a - \lambda/2, 0)$, which proves Equation (8). The stated cue reconstruction error follows immediately.

**Proof of Lemma B.2.** Let $\boldsymbol{x} \coloneqq \mathbf{W}^\top \boldsymbol{h} = (x_1, \ldots, x_k)$. As in the proof of Theorem B.4, Top-$K$ decoding in an orthonormal basis selects a support of size at most $K$ that maximizes $\sum_{i \in S} x_i^2$, and then sets $z_i^{(K)} = x_i$ on that support. If $a^2 = x_1^2$ is at least the $K$-th largest value among $\{x_2^2, \ldots, x_k^2\}$, then index 1 belongs to the set of $K$ largest values among $\{x_1^2, \ldots, x_k^2\}$, and Top-$K$ must select index 1. Thus $z_1^{(K)} = x_1 = a$, proving Equation (9). For the probability bound, the failure event $\{z_1^{(K)} \neq a\}$ implies $\max_{2 \leq i \leq k} x_i^2 > a^2$, hence $\max_{2 \leq i \leq k} |x_i| > a$. By Equation (5), for $i \geq 2$ the random variables $x_i$ are i.i.d. $\mathcal{N}(0, \sigma^2)$ with $\sigma^2 = b^2/(k-1)$. A union bound and the Gaussian tail inequality yield

$$\mathbb{P}\left( \max_{2 \leq i \leq k} |x_i| > a \right) \leq \sum_{i=2}^k \mathbb{P}\left( |x_i| > a \right) = (k-1)\mathbb{P}\left( |X| > a \right) \leq 2(k-1)\exp\left( -\frac{a^2}{2\sigma^2} \right),$$

which is Equation (10).

**Proof of Theorem B.3.** We proceed in steps.

**Step 1: reduce the optimization to coordinates.** Since $\mathbf{W}$ has orthonormal columns, for any $\boldsymbol{z} \in \mathbb{R}^k$,

$$\|\boldsymbol{h} - \mathbf{W}\boldsymbol{z}\|_2^2 = \|\mathbf{W}^\top\boldsymbol{h} - \mathbf{W}^\top\mathbf{W}\boldsymbol{z}\|_2^2 + \|\boldsymbol{h} - \mathbf{W}\mathbf{W}^\top\boldsymbol{h}\|_2^2 = \|\boldsymbol{x} - \boldsymbol{z}\|_2^2 + C_0,$$

where $\boldsymbol{x} \coloneqq \mathbf{W}^\top\boldsymbol{h}$ and $C_0 \coloneqq \|\boldsymbol{h} - \mathbf{W}\mathbf{W}^\top\boldsymbol{h}\|_2^2$ does not depend on $\boldsymbol{z}$. Therefore, minimizing Equation (6) is equivalent to

$$\min_{\boldsymbol{z} \in \mathbb{R}^k} \|\boldsymbol{x} - \boldsymbol{z}\|_2^2 + \lambda\|\boldsymbol{z}\|_1. \tag{15}$$

**Step 2: separability and the soft-thresholding solution.** The objective in Equation (15) is separable across coordinates:

$$\|\boldsymbol{x} - \boldsymbol{z}\|_2^2 + \lambda\|\boldsymbol{z}\|_1 = \sum_{i=1}^k \left( (x_i - z_i)^2 + \lambda|z_i| \right).$$

Thus a minimizer $\boldsymbol{z}^\star$ can be obtained by minimizing each coordinate independently:

$$z_i^\star \in \arg\min_{z \in \mathbb{R}} (x_i - z)^2 + \lambda|z|.$$

We claim that the unique minimizer is $z_i^\star = \mathrm{ST}(x_i, \lambda/2)$. To verify, fix $x \in \mathbb{R}$ and consider $f(z) \coloneqq (x - z)^2 + \lambda|z|$.

If $z > 0$, then $f(z) = (x - z)^2 + \lambda z$ and $f'(z) = 2(z - x) + \lambda$. Setting $f'(z) = 0$ yields $z = x - \lambda/2$. This solution lies in the region $z > 0$ iff $x > \lambda/2$.

If $z < 0$, then $f(z) = (x - z)^2 - \lambda z$ and $f'(z) = 2(z - x) - \lambda$. Setting $f'(z) = 0$ yields $z = x + \lambda/2$. This solution lies in the region $z < 0$ iff $x < -\lambda/2$.

If $|x| \leq \lambda/2$, then the stationary points above fall outside their regions. In this case, the minimizer occurs at the non-differentiable point $z = 0$. To confirm, note that for $z > 0$ with $|x| \leq \lambda/2$,

$$f'(z) = 2(z - x) + \lambda \geq 2(0 - |x|) + \lambda \geq 0,$$

so $f$ is non-decreasing on $(0, \infty)$ and minimized at $z = 0$. Similarly for $z < 0$,

$$f'(z) = 2(z - x) - \lambda \leq 2(0 + |x|) - \lambda \leq 0,$$

so $f$ is non-increasing on $(-\infty, 0)$ and minimized at $z = 0$. Therefore,

$$z^\star = \mathrm{ST}(x, \lambda/2) = \mathrm{sign}(x)\max\left( |x| - \lambda/2, 0 \right).$$

Applying this to each coordinate yields $z_i^\star = \mathrm{ST}(x_i, \lambda/2)$.

**Step 3: compute $\mathbb{E}[\|z_{2:k}^\star\|_2^2]$ in terms of a one-dimensional expectation.** By Equation (5), $x_2, \ldots, x_k$ are i.i.d. $\mathcal{N}(0, \sigma^2)$ with $\sigma^2 = \frac{b^2}{k-1}$. By Step 2,

$$z_i^\star = \mathrm{ST}(x_i, \lambda/2) \quad \text{for} \quad i \in \{2, \ldots, k\}.$$

Thus

$$\mathbb{E}\left[\|\boldsymbol{z}_{2:k}^\star\|_2^2\right] = \sum_{i=2}^{k} \mathbb{E}\left[(z_i^\star)^2\right] = (k-1)\mathbb{E}\left[\mathrm{ST}(X, \lambda/2)^2\right], \tag{16}$$

where $X \sim \mathcal{N}(0, \sigma^2)$.

**Step 4: compute $\mathbb{E}[\mathrm{ST}(X, t)^2]$ in closed form.** Let $t > 0$ and set $u := t/\sigma$. Write $X = \sigma Z$ with $Z \sim \mathcal{N}(0, 1)$. Then

$$\mathrm{ST}(X, t) = \mathrm{sign}(Z)\sigma \max\left(|Z| - u, 0\right), \qquad \mathrm{ST}(X, t)^2 = \sigma^2\left(|Z| - u\right)^2 \mathbb{1}[|Z| > u].$$

By symmetry,

$$\mathbb{E}\left[\mathrm{ST}(X, t)^2\right] = 2\sigma^2 \int_u^\infty (z - u)^2 \phi(z)\, \mathrm{d}z. \tag{17}$$

Expand $(z - u)^2 = z^2 - 2uz + u^2$ and integrate term by term:

$$\int_u^\infty (z - u)^2 \phi(z)\, \mathrm{d}z = \int_u^\infty z^2 \phi(z)\, \mathrm{d}z - 2u \int_u^\infty z \phi(z)\, \mathrm{d}z + u^2 \int_u^\infty \phi(z)\, \mathrm{d}z. \tag{18}$$

We now evaluate each integral. First,

$$\int_u^\infty z\phi(z)\, \mathrm{d}z = \phi(u), \tag{19}$$

since $\frac{\mathrm{d}}{\mathrm{d}z}(-\phi(z)) = z\phi(z)$. Second,

$$\int_u^\infty \phi(z)\, \mathrm{d}z = 1 - \Phi(u). \tag{20}$$

Third, using $\phi'(z) = -z\phi(z)$,

$$\int_u^\infty z^2 \phi(z)\, \mathrm{d}z = \int_u^\infty z(-\phi'(z))\, \mathrm{d}z = -\left[z\phi(z)\right]_u^\infty + \int_u^\infty \phi(z)\, \mathrm{d}z = u\phi(u) + 1 - \Phi(u). \tag{21}$$

Substituting Equations (19) to (21) into Equation (18) gives

$$\int_u^\infty (z - u)^2 \phi(z)\, \mathrm{d}z = \left(u\phi(u) + 1 - \Phi(u)\right) - 2u\phi(u) + u^2\left(1 - \Phi(u)\right) = (1 + u^2)\left(1 - \Phi(u)\right) - u\phi(u).$$

Combining with Equation (17) yields

$$\mathbb{E}\left[\mathrm{ST}(X, t)^2\right] = 2\sigma^2\left((1 + u^2)(1 - \Phi(u)) - u\phi(u)\right). \tag{22}$$

**Step 5: upper bound using Mill's ratio and define $\Psi$.** We use the standard inequality for $u > 0$,

$$1 - \Phi(u) \leq \frac{\phi(u)}{u}. \tag{23}$$

Substitute Equation (23) into Equation (22):

$$\mathbb{E}\left[\mathrm{ST}(X, t)^2\right] \leq 2\sigma^2\left((1 + u^2)\frac{\phi(u)}{u} - u\phi(u)\right) = 2\sigma^2\left(\frac{\phi(u)}{u} + u\phi(u) - u\phi(u)\right) = 2\sigma^2 \frac{\phi(u)}{u}. \tag{24}$$

We now set $t = \lambda/2$, so $u = \frac{\lambda}{2\sigma} = \frac{\lambda\sqrt{k-1}}{2b}$. Plugging Equation (24) into Equation (16) gives

$$\mathbb{E}\left[\|\boldsymbol{z}_{2:k}^\star\|_2^2\right] \leq (k-1)\left(2\sigma^2 \frac{\phi(u)}{u}\right) = 2b^2 \frac{\phi(u)}{u} = b^2 \Psi(u),$$

where we define $\Psi(u) := \frac{2\phi(u)}{u}$ as in Equation (11). Finally, since $\phi(u) = \frac{1}{\sqrt{2\pi}} \exp\left(-\frac{u^2}{2}\right)$, we obtain Equation (12). This concludes the proof.

**Proof of Theorem B.4.** We prove Equation (13) by reducing Top-$K$ decoding in an orthonormal basis to a bound on the maximum of Gaussian coordinates and then integrating a tail inequality.

**Step 1: Top-$K$ decoding in an orthonormal basis.** Let $\boldsymbol{x} := \mathbf{W}^\top \boldsymbol{h} \in \mathbb{R}^k$. Since $\mathbf{W}$ has orthonormal columns, for any $\boldsymbol{z} \in \mathbb{R}^k$,

$$\|\boldsymbol{h} - \mathbf{W}\boldsymbol{z}\|_2^2 = \|\mathbf{W}^\top\boldsymbol{h} - \boldsymbol{z}\|_2^2 + \|(\mathbf{I} - \mathbf{W}\mathbf{W}^\top)\boldsymbol{h}\|_2^2.$$

The second term is independent of $\boldsymbol{z}$, hence $\boldsymbol{z}^{(K)}$ minimizes $\|\boldsymbol{x} - \boldsymbol{z}\|_2^2$ subject to $\|\boldsymbol{z}\|_0 \leq K$. For any fixed support $S \subseteq \{1, \ldots, k\}$ with $|S| \leq K$, the minimizer is $z_i = x_i$ for $i \in S$ and $z_i = 0$ otherwise, with objective value $\sum_{i \notin S} x_i^2$. Therefore, an optimal support maximizes $\sum_{i \in S} x_i^2$, so Top-$K$ selects indices of the $K$ largest values of $x_i^2$.

**Step 2: reduction to residual coordinates and a monotone upper bound.** Let $x_i := \langle \boldsymbol{v}_i, \boldsymbol{h} \rangle$ so that $\boldsymbol{x} = (x_1, \ldots, x_k)$. By Equation (5), the residual coordinates $(x_2, \ldots, x_k)$ are i.i.d. $\mathcal{N}(0, \sigma^2)$ with $\sigma^2 = \frac{b^2}{k-1}$. Since $R_K$ keeps at most $K$ coordinates from $(x_2, \ldots, x_k)$, it is upper bounded by the sum of the $K$ largest squared residual coordinates:

$$R_K \leq \sum_{j=1}^K X_{(j)}^2, \tag{25}$$

where $X_{(1)}^2 \geq \cdots \geq X_{(k-1)}^2$ are the order statistics of $X_1^2, \ldots, X_{k-1}^2$ for i.i.d. $X_i \sim \mathcal{N}(0, \sigma^2)$.

**Step 3: upper bound by $K$ times the maximum.** Let

$$M := \max_{1 \leq i \leq k-1} X_i^2.$$

Since $X_{(j)}^2 \leq M$ for all $j$, we have

$$\sum_{j=1}^K X_{(j)}^2 \leq KM. \tag{26}$$

Combining Equations (25) and (26) and taking expectation yields

$$\mathbb{E}[R_K] \leq K\mathbb{E}[M]. \tag{27}$$

**Step 4: a tail bound for the maximum.** Let $Y := \max_{1 \leq i \leq k-1} |X_i|$, so $M = Y^2$. For any $t \geq 0$, by a union bound and symmetry,

$$\mathbb{P}(Y \geq t) \leq \sum_{i=1}^{k-1} \mathbb{P}(|X_i| \geq t) = (k-1)\mathbb{P}(|X| \geq t),$$

where $X \sim \mathcal{N}(0, \sigma^2)$. Write $X = \sigma Z$ with $Z \sim \mathcal{N}(0, 1)$. Using $\mathbb{P}(|Z| \geq u) \leq 2\exp(-u^2/2)$ for $u \geq 0$, we obtain

$$\mathbb{P}(|X| \geq t) = \mathbb{P}(|Z| \geq t/\sigma) \leq 2\exp\left(-\frac{t^2}{2\sigma^2}\right),$$

and therefore

$$\mathbb{P}(Y \geq t) \leq 2(k-1)\exp\left(-\frac{t^2}{2\sigma^2}\right). \tag{28}$$

**Step 5: bounding $\mathbb{E}[M]$ by tail integration.** Since $Y \geq 0$, the tail integral identity gives

$$\mathbb{E}[Y^2] = \int_0^\infty \mathbb{P}(Y^2 \geq u)\, \mathrm{d}u = \int_0^\infty \mathbb{P}(Y \geq \sqrt{u})\, \mathrm{d}u.$$

Using Equation (28),

$$\mathbb{E}[Y^2] \leq \int_0^\infty \min\left(1, 2(k-1)\exp\left(-\frac{u}{2\sigma^2}\right)\right)\, \mathrm{d}u.$$

Let $u_0 := 2\sigma^2 \log(2(k-1))$ so that $2(k-1)\exp(-u_0/(2\sigma^2)) = 1$. Splitting the integral at $u_0$ yields

$$
\begin{aligned}
\mathbb{E}[Y^2] &\leq \int_0^{u_0} 1\, \mathrm{d}u + \int_{u_0}^{\infty} 2(k-1)\exp\left(-\frac{u}{2\sigma^2}\right)\mathrm{d}u \\
&= u_0 + 4\sigma^2(k-1)\exp\left(-\frac{u_0}{2\sigma^2}\right) \\
&= 2\sigma^2 \log(2(k-1)) + 2\sigma^2.
\end{aligned}
\tag{29}
$$

Because $M = Y^2$, Equation (29) implies

$$
\mathbb{E}[M] \leq 2\sigma^2 \log(2(k-1)) + 2\sigma^2.
\tag{30}
$$

**Step 6: conclude the bound on $\mathbb{E}[R_K]$.**   Substituting Equation (30) into Equation (27) gives

$$
\mathbb{E}[R_K] \leq K\left(2\sigma^2 \log(2(k-1)) + 2\sigma^2\right) = \sigma^2 K\left(2\log(2(k-1)) + 2\right),
$$

which is Equation (13). Dividing by $b^2 = (k-1)\sigma^2$ yields Equation (14). This completes the proof.

# C. Experiment results on additional models

To assess the generality of our findings beyond the Gemma-3 family, we apply the same experimental protocol described in Sections 4 and 5 to three additional models: Llama-3.1-8B (Grattafiori et al., 2024) and two models from the Gemma-2 family (Gemma-2-9B and Gemma-2-2B) (Gemma Team et al., 2024). These models span different architectural families and scales, providing a comprehensive test of robustness across contemporary open-weight language models.

## C.1. Llama-3.1-8B

We analyze Llama-3.1-8B, an 8B parameter model with 32 transformer layers from Meta AI. We focus on layer 16 (50% depth). We use sparse autoencoders from the Llama Scope release (He et al., 2024) with 32,768 features trained on residual stream activations.

Table 6 summarizes feature detection and token injection results for both reasoning datasets. As in the main experiments, we identify the top 100 features by Cohen's $d$ and classify them using the token injection framework from Section 4.3.

*Table 6.* Feature detection and token injection results for Llama-3.1-8B. TD denotes token-driven, PTD partially token-driven, WTD weakly token-driven, and CD context-dependent.

| Dataset | Mean $d$ | TD | PTD | WTD | CD | Avg. Inject. $d$ |
|---|---|---|---|---|---|---|
| s1K | 0.851 | 37 (37%) | 18 (18%) | 28 (28%) | 17 (17%) | 0.909 |
| General Inquiry CoT | 1.015 | 26 (26%) | 17 (17%) | 24 (24%) | 33 (33%) | 0.613 |

Llama-3.1-8B exhibits moderate token injection effects intermediate between Gemma-3 and Gemma-2 models. For s1K, 83% of features show some degree of token-driven behavior (combining all three categories), with an average injection Cohen's $d$ of 0.909. For General Inquiry CoT, 67% show token-driven behavior with average $d$ of 0.613. The higher proportion of context-dependent features on General Inquiry CoT (33%) compared to s1K (17%) mirrors patterns observed in other models, suggesting dataset-specific differences in feature activation patterns.

To investigate the context-dependent features, we apply the LLM-guided falsification protocol from Section 4.4 to a random sample from each dataset (17 features from s1K, 20 from General Inquiry CoT). Across all 37 analyzed features, none satisfy the criteria for genuine reasoning behavior. The LLM classifies all features as confounds, with high confidence for 30 of them (81%). The dominant confounds include conversational discourse markers, formal writing style, meta-cognitive planning phrases, and technical vocabulary that appears in both reasoning and non-reasoning contexts.

## C.2. Gemma-2-9B

We analyze Gemma-2-9B, a 9B parameter model with 42 transformer layers, focusing on layer 21 (50% depth). We use sparse autoencoders from the GemmaScope release (Lieberum et al., 2024) with 16,384 features trained on residual stream activations.

Table 7 summarizes feature detection and token injection results for both reasoning datasets. As in the main experiments, we identify the top 100 features by Cohen's $d$ and classify them using the token injection framework from Section 4.3.

*Table 7.* Feature detection and token injection results for Gemma-2-9B. TD denotes token-driven, PTD partially token-driven, WTD weakly token-driven, and CD context-dependent.

| Dataset | Mean $d$ | TD | PTD | WTD | CD | Avg. Inject. $d$ |
|---|---|---|---|---|---|---|
| s1K | 0.709 | 6 (8%) | 7 (9%) | 23 (29%) | 42 (54%) | 0.285 |
| General Inquiry CoT | 0.764 | 6 (6%) | 4 (4%) | 33 (33%) | 57 (57%) | 0.294 |

Compared to Gemma-3 models, Gemma-2-9B exhibits a substantially larger fraction of context-dependent features and markedly smaller injection effect sizes. Although statistically significant activation differences between reasoning and non-reasoning text are still observed at the detection stage, the average Cohen's $d$ induced by token injection is below 0.3 for both datasets. This indicates weaker reliance on shallow token-level cues, but does not by itself imply the presence of genuine reasoning features.

To further investigate these context-dependent features, we apply the LLM-guided falsification protocol from Section 4.4 to 20 randomly sampled features per dataset. Across all 40 analyzed features, none satisfy the criteria for genuine reasoning behavior. The LLM classifies all features as confounds, with high confidence for 34 of them. The dominant confounds include formal academic writing style, meta-cognitive discourse markers, and technical vocabulary that appears in both reasoning and non-reasoning contexts.

### C.3. Gemma-2-2B

We next analyze Gemma-2-2B, a 2B parameter model with 26 layers, using the same methodology. We again select layer 13 (50% depth) and apply an SAE with 16,384 features.

Table 8 reports the corresponding feature detection and token injection results.

*Table 8.* Feature detection and token injection results for Gemma-2-2B.

| Dataset | Mean $d$ | TD | PTD | WTD | CD | Avg. Inject. $d$ |
|---|---|---|---|---|---|---|
| s1K | 0.636 | 2 (2%) | 18 (18%) | 36 (36%) | 44 (44%) | 0.281 |
| General Inquiry CoT | 0.602 | 1 (1%) | 7 (7%) | 35 (35%) | 57 (57%) | 0.207 |

Gemma-2-2B shows the weakest token injection effects among all models studied. Only a small fraction of features are classified as strongly token-driven, while between 44% and 57% are context-dependent. Average injection effect sizes are also the lowest observed. Despite this reduced token sensitivity, LLM-guided analysis of 40 context-dependent features again identifies no genuine reasoning features. Thirty-six of these features are classified as confounds with high confidence. The identified confounds include procedural discourse markers, formal sentence structure, and domain-specific vocabulary patterns.

# D. Experiment results on alternative ranking metrics

In the main text, we use Cohen's $d$ as the primary criterion for ranking and selecting candidate reasoning features (Section 4.2). To evaluate the robustness of our conclusions to this choice, we repeat the full experimental pipeline using two alternative ranking metrics: ROC-AUC and frequency ratio. For every configuration, we rank all SAE features by the corresponding metric and select the top 100 features that satisfy the following criteria: minimum Cohen's $d$ effect size $d \geq 0.3$, Bonferroni-corrected $p$-value $\leq 0.01$ under the Mann-Whitney $U$ test, and ROC-AUC $\geq 0.6$ when using the feature activation as a univariate classifier. All experiments in this section are conducted on Gemma-3-4B-Instruct at layer 22 using the s1K dataset, with all other parameters held fixed, including datasets, sample sizes, statistical thresholds, token injection strategies, and LLM-guided falsification.

## D.1. ROC-AUC based selection

ROC-AUC measures a feature's ability to discriminate between reasoning and non-reasoning samples across all possible thresholds. For a feature $i$, it is defined as

$$\text{AUC}_i = \mathbb{P}\left(a_{i,j}^{\text{R}} > a_{i,k}^{\text{NR}}\right) = \frac{1}{n_{\text{R}} n_{\text{NR}}} \sum_{j=1}^{n_{\text{R}}} \sum_{k=1}^{n_{\text{NR}}} \mathbb{1}\left[a_{i,j}^{\text{R}} > a_{i,k}^{\text{NR}}\right].$$

This metric is distribution-free, invariant to monotonic transformations, and robust to class imbalance. We rank all SAE features by ROC-AUC and select the top 100 features satisfying AUC $\geq 0.6$, Bonferroni-corrected $p \leq 0.01$, and Cohen's $d \geq 0.3$.

Table 9 summarizes the resulting feature statistics and injection outcomes. Features selected by ROC-AUC exhibit token injection behavior that closely mirrors the results obtained using Cohen's $d$ ranking. In particular, 95% of features fall into one of the token-driven categories, while only a small fraction remain context-dependent. The mean Cohen's $d$ of the selected features is comparable to that of the Cohen's $d$ ranked set, indicating strong agreement between the two metrics.

*Table 9.* Results for the top 100 features ranked by ROC-AUC on Gemma-3-4B-Instruct at layer 22 using the s1K dataset.

| Metric | Value |
|---|---|
| Mean Cohen's $d$ | 0.830 |
| Mean ROC-AUC | 0.667 |
| Token-driven | 59 (59%) |
| Partially token-driven | 18 (18%) |
| Weakly token-driven | 18 (18%) |
| Context-dependent | 5 (5%) |
| Features analyzed (LLM) | 5 |
| Genuine reasoning features | 0 |

All five context-dependent features identified under ROC-AUC ranking are classified as confounds by the LLM-guided falsification procedure described in Section 4.4, with high confidence in each case.

## D.2. Frequency ratio based selection

We next consider the frequency ratio metric, which measures how much more often a feature activates on reasoning samples than on non-reasoning samples. For feature $i$, we define

$$\text{FreqRatio}_i = \frac{\text{freq}_i^{\text{R}} + \epsilon}{\text{freq}_i^{\text{NR}} + \epsilon},$$

where $\text{freq}_i^{\text{R}}$ and $\text{freq}_i^{\text{NR}}$ denote the proportion of samples whose maximum activation exceeds a fixed threshold, and $\epsilon = 0.01$ is a smoothing constant. The activation threshold is set to $\max(0.5 \cdot \sigma_{\text{baseline}}, 0.01)$, following the same procedure used in the auxiliary analyses described in Section 4.2.

Ranking features by frequency ratio and selecting the top 100 yields results that are nearly identical to those obtained with ROC-AUC. As shown in Table 10, 96% of selected features exhibit statistically significant token-driven behavior, with

only four features classified as context-dependent. These features also show comparable effect sizes and discriminative performance.

*Table 10.* Results for the top 100 features ranked by frequency ratio on Gemma-3-4B-Instruct at layer 22 using the s1K dataset.

| Metric | Value |
|---|---|
| Mean Cohen's $d$ | 0.830 |
| Mean ROC-AUC | 0.667 |
| Token-driven | 65 (65%) |
| Partially token-driven | 14 (14%) |
| Weakly token-driven | 17 (17%) |
| Context-dependent | 4 (4%) |
| Features analyzed (LLM) | 4 |
| Genuine reasoning features | 0 |

LLM-guided analysis of all four context-dependent features again identifies no genuine reasoning features, with each classified as a confound.

### D.3. Comparison across ranking metrics

To quantify overlap between feature sets selected by different ranking metrics, we compute Jaccard similarities between the top 100 features selected by Cohen's $d$, ROC-AUC, and frequency ratio. Figure 6 shows that the overlap is substantial across all pairs, indicating that the three metrics largely prioritize the same subset of features.

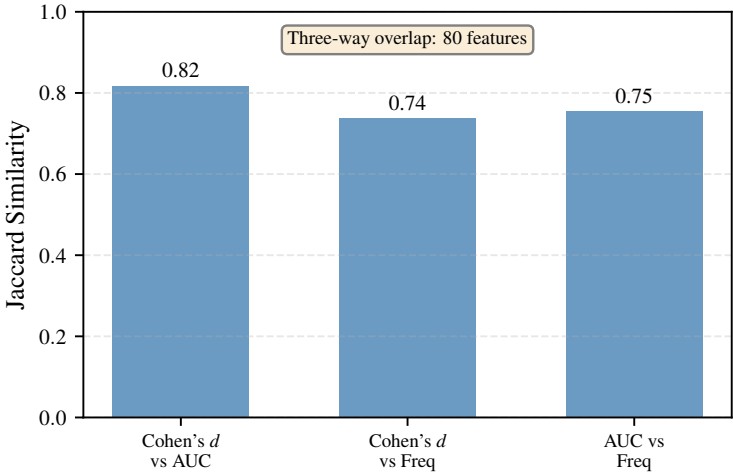

*Figure 6.* Jaccard similarity between top-100 feature sets selected by different ranking metrics on Gemma-3-4B-Instruct at layer 22 using the s1K dataset.

Taken together, these results demonstrate that our main conclusions do not depend on the specific choice of ranking metric. Whether features are selected by Cohen's $d$, ROC-AUC, or frequency ratio, the vast majority are explained by token-level or short-range contextual confounds, and no genuine reasoning features are identified under any selection criterion.

# E. Additional experimental statistics

This section reports supplementary statistics from the main experiments that provide additional quantitative context for interpreting the results presented in Section 5.

## E.1. Token dependency statistics across configurations

Beyond the categorical classification induced by token injection, we analyze the degree to which feature activations are concentrated on a small subset of tokens. For each feature, we compute a token concentration score defined as the fraction of total activation mass attributable to the top 30 tokens ranked by mean activation. We additionally report normalized activation entropy as a complementary measure of dispersion. High concentration and low entropy indicate strong reliance on a limited set of lexical triggers.

Table 11 summarizes token dependency statistics for all main configurations. Across models, layers, and datasets, between 30% and 78% of features exhibit high token dependency, defined as concentration greater than 0.5. Notably, s1K configurations generally show lower token concentration than General Inquiry CoT configurations for the same model and layer. Gemma-3-4B-Instruct layer 17 on s1K shows the lowest token concentration (mean: 0.402, median: 0.265), while Gemma-3-12B-Instruct layer 27 on General Inquiry CoT shows the highest (mean: 0.714, median: 0.735), suggesting both architectural and dataset-specific influences on token dependency.

*Table 11.* Token concentration statistics for top-ranked features across all main configurations. High dependency denotes features with concentration greater than 0.5.

| Model | Layer | Dataset | Mean | Median | High dep. |
|---|---|---|---|---|---|
| Gemma-3-12B-Instruct | 17 | s1K | 0.529 | 0.478 | 41 (41%) |
| Gemma-3-12B-Instruct | 17 | Gen. Inq. | 0.641 | 0.685 | 73 (73%) |
| Gemma-3-12B-Instruct | 22 | s1K | 0.461 | 0.457 | 30 (30%) |
| Gemma-3-12B-Instruct | 22 | Gen. Inq. | 0.622 | 0.681 | 73 (73%) |
| Gemma-3-12B-Instruct | 27 | s1K | 0.521 | 0.480 | 41 (41%) |
| Gemma-3-12B-Instruct | 27 | Gen. Inq. | 0.714 | 0.735 | 78 (78%) |
| Gemma-3-4B-Instruct | 17 | s1K | 0.402 | 0.265 | 32 (32%) |
| Gemma-3-4B-Instruct | 17 | Gen. Inq. | 0.639 | 0.699 | 63 (63%) |
| Gemma-3-4B-Instruct | 22 | s1K | 0.539 | 0.449 | 47 (47%) |
| Gemma-3-4B-Instruct | 22 | Gen. Inq. | 0.690 | 0.836 | 69 (69%) |
| Gemma-3-4B-Instruct | 27 | s1K | 0.528 | 0.473 | 48 (48%) |
| Gemma-3-4B-Instruct | 27 | Gen. Inq. | 0.674 | 0.760 | 71 (71%) |
| DS-R1-Distill-Llama-8B | 19 | s1K | 0.569 | 0.528 | 52 (52%) |
| DS-R1-Distill-Llama-8B | 19 | Gen. Inq. | 0.687 | 0.826 | 69 (69%) |

For s1K configurations, median concentration values are often lower than means, indicating left-skewed distributions where most features have moderate-to-low concentration with a minority showing very high concentration. Conversely, General Inquiry CoT configurations show median values close to or higher than means, suggesting more uniform high concentration. Figure 7 visualizes the distribution of token concentration across layers for representative models.

## E.2. Injection strategy performance

We further analyze which injection strategies are most effective at eliciting feature activation. For a representative configuration, Gemma-3-12B-Instruct at layer 22 on s1K, we identify the injection strategy that yields the largest Cohen's $d$ for each feature. The prepend strategy dominates, accounting for 69.7% of best-performing cases, followed by bigram injection, replacement, and trigram injection. Context-preserving strategies collectively account for a small fraction of cases.

This distribution indicates that simple token presence is typically sufficient to drive feature activation, and that preserving local co-occurrence or syntactic context does not substantially attenuate injection effects. Figure 8 compares Cohen's $d$ distributions across strategies.

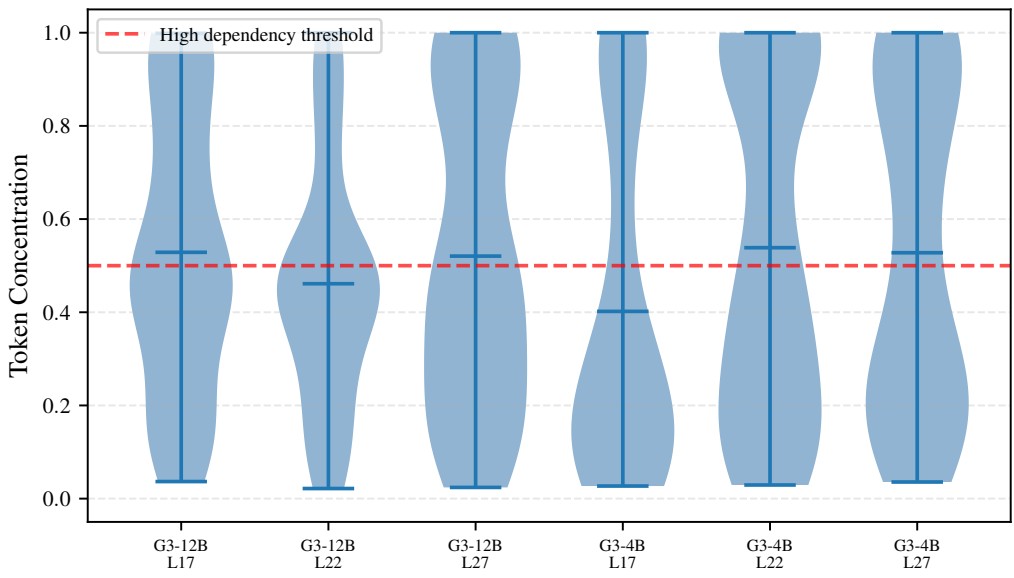

*Figure 7.* Token concentration distributions across layers for Gemma-3-12B-Instruct and Gemma-3-4B-Instruct on the s1K dataset.

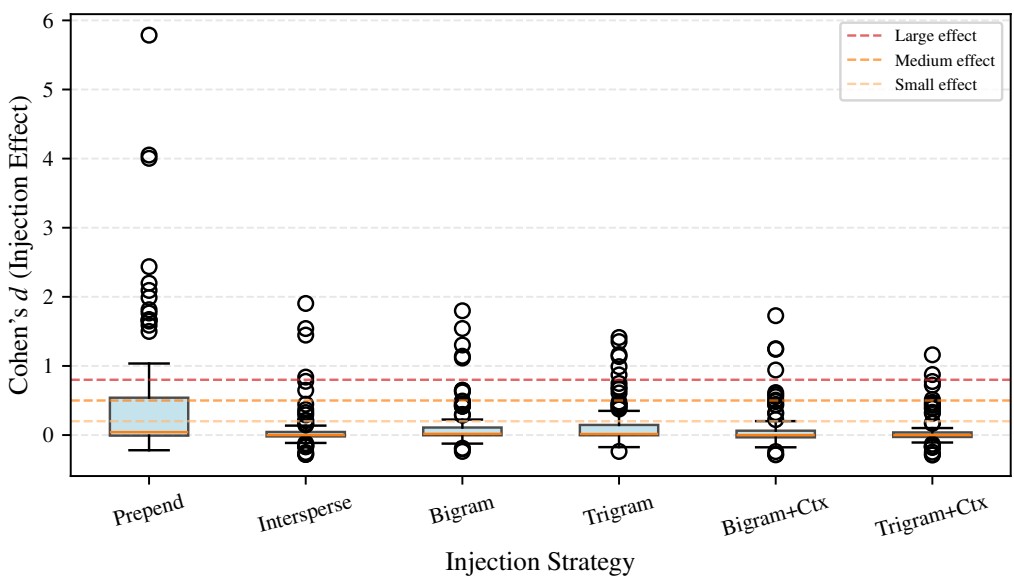

*Figure 8.* Cohen's $d$ distributions across token injection strategies for Gemma-3-12B-Instruct at layer 22 on s1K.

### E.3. Average injection effect sizes across configurations

This section reports the magnitude of activation changes induced by token injection across all experimental configurations. For each model, layer, and reasoning dataset, we compute the average Cohen's $d$ achieved by the best-performing injection strategy for each of the 100 candidate features, as defined in Section 4.3. The reported values therefore summarize the strength of the strongest token-level intervention per feature, averaged across features within each configuration.

Table 12 presents the resulting average effect sizes. Across all 14 configurations, the mean Cohen's $d$ ranges from 0.521 to 1.471, with an overall mean of 0.974 and median of 0.982. All configurations exceed $d = 0.5$, which corresponds to a medium effect size under standard conventions (Cohen, 2013), and 6 of the 14 configurations exceed $d = 1.0$, indicating very large effects.

*Table 12.* Average Cohen's $d$ for token injection across configurations. Values are computed by averaging, for each configuration, the best injection effect size per feature across the top 100 candidate features.

| Model | Layer | s1K | General Inquiry CoT |
|---|---|---|---|
| Gemma-3-12B-Instruct | 17 | 1.266 | 0.982 |
| Gemma-3-12B-Instruct | 22 | 0.539 | 0.660 |
| Gemma-3-12B-Instruct | 27 | 1.044 | 0.698 |
| Gemma-3-4B-Instruct | 17 | 1.404 | 1.078 |
| Gemma-3-4B-Instruct | 22 | 1.409 | 0.913 |
| Gemma-3-4B-Instruct | 27 | 1.471 | 0.756 |
| DeepSeek-R1-Distill-Llama-8B | 19 | 0.899 | 0.521 |

These effect sizes indicate that injecting a small number of feature-associated tokens into non-reasoning text is sufficient to induce substantial activation shifts. In particular, injecting three tokens into 64-token sequences, which corresponds to approximately 4.7% of the input tokens, produces activation increases that are comparable to or exceed conventional medium and large effect size thresholds. The consistency of these effects across models, layers, and datasets provides strong evidence that token-level patterns play a dominant role in driving the activation of features identified by contrastive methods.

We observe systematic variation across configurations. Gemma-3 models generally exhibit larger average injection effects than DeepSeek-R1-Distill-Llama-8B. Gemma-3-4B-Instruct shows the highest injection effects (1.404–1.471 on s1K), while Gemma-3-12B-Instruct layer 22 shows the lowest among Gemma-3 models (0.539–0.660). Nevertheless, even the smallest observed average effect size represents a medium effect, underscoring that token injection reliably induces strong feature activation across all tested conditions.

### E.4. Activation magnitude analysis

To contextualize effect sizes, we analyze absolute activation magnitudes across conditions. For each feature, we compute the mean activation across samples within each condition and then average these values across features.

Table 13 reports activation statistics for Gemma-3-12B-Instruct at layer 22 on s1K. Token injection increases mean activation by a factor of approximately 1.32 relative to the non-reasoning baseline and reaches parity with activations observed on genuine reasoning text. The overlap in upper-tail statistics further indicates that token injection is sufficient to reproduce the activation regimes associated with reasoning inputs.

*Table 13.* Activation magnitude statistics for Gemma-3-12B-Instruct at layer 22 on s1K. Injected refers to the best-performing token injection strategy per feature.

| Condition | Mean | Std. | Median | 90th pct. |
|---|---|---|---|---|
| Non-reasoning (baseline) | 416.5 | 1102.8 | 26.2 | 1366.8 |
| Non-reasoning (injected) | 549.8 | 1123.1 | 68.5 | 1971.5 |
| Reasoning text | 558.3 | 1139.5 | 144.2 | 2010.3 |

### E.5. LLM-guided falsification convergence

We analyze the convergence behavior of the LLM-guided falsification protocol described in Section 4.4. Table 14 reports the number of iterations required to reach the stopping criteria across all 153 context-dependent features from the main experiments.

*Table 14.* Iteration statistics for LLM-guided falsification across all analyzed context-dependent features.

| Statistic | Value |
|---|---|
| Mean iterations to convergence | 2.4 |
| Median iterations to convergence | 2.0 |
| Converged in 1 iteration | 47 (31%) |
| Converged in 2 iterations | 68 (44%) |
| Converged in 3 or more iterations | 38 (25%) |
| Reached max iterations (10) | 3 (2%) |

Most features converge rapidly, with 75% resolving within two iterations. Features requiring more iterations typically involve overlapping stylistic or discourse-level patterns.

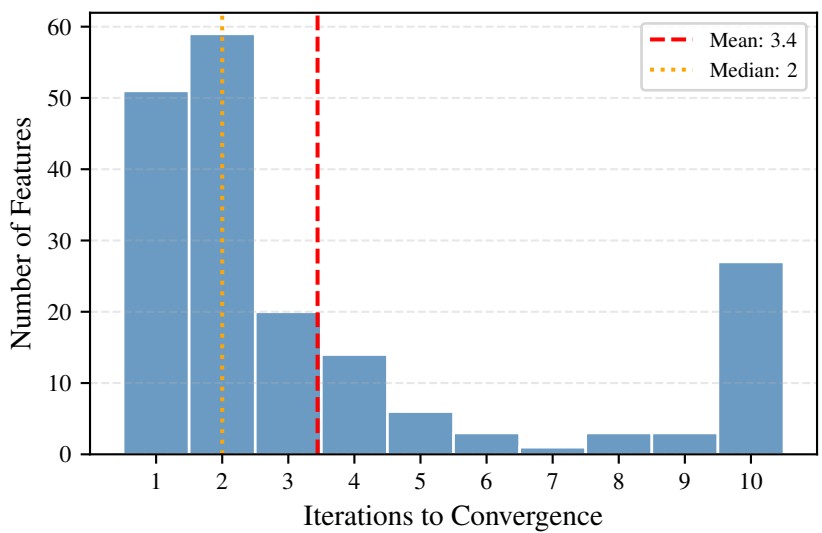

*Figure 9.* Distribution of iterations required for LLM-guided falsification to converge.

### E.6. Feature overlap across reasoning datasets

Finally, we examine whether the same features are selected when ranking on different reasoning datasets. For each model and layer, we compute the Jaccard similarity between the top 100 features identified using s1K and General Inquiry CoT.

Table 15 shows that overlap is generally low, with Jaccard similarities ranging from 0.058 to 0.190. Larger models exhibit moderately higher overlap, while deeper layers tend to show reduced overlap. These results indicate that different reasoning datasets activate largely distinct feature subsets.

Figure 10 provides a visual summary of feature overlap across datasets. LLM-guided analysis of shared context-dependent features again identifies only confounds, including mathematical notation detectors, formal discourse markers, and meta-cognitive phrases that appear across reasoning and non-reasoning text.

*Table 15.* Overlap between top-ranked features selected using s1K and General Inquiry CoT.

| Model | Layer | Intersection | Jaccard |
|---|---|---|---|
| Gemma-3-12B-Instruct | 17 | 27 | 0.156 |
| Gemma-3-12B-Instruct | 22 | 32 | 0.190 |
| Gemma-3-12B-Instruct | 27 | 12 | 0.064 |
| Gemma-3-4B-Instruct | 17 | 11 | 0.058 |
| Gemma-3-4B-Instruct | 22 | 16 | 0.087 |
| Gemma-3-4B-Instruct | 27 | 13 | 0.070 |

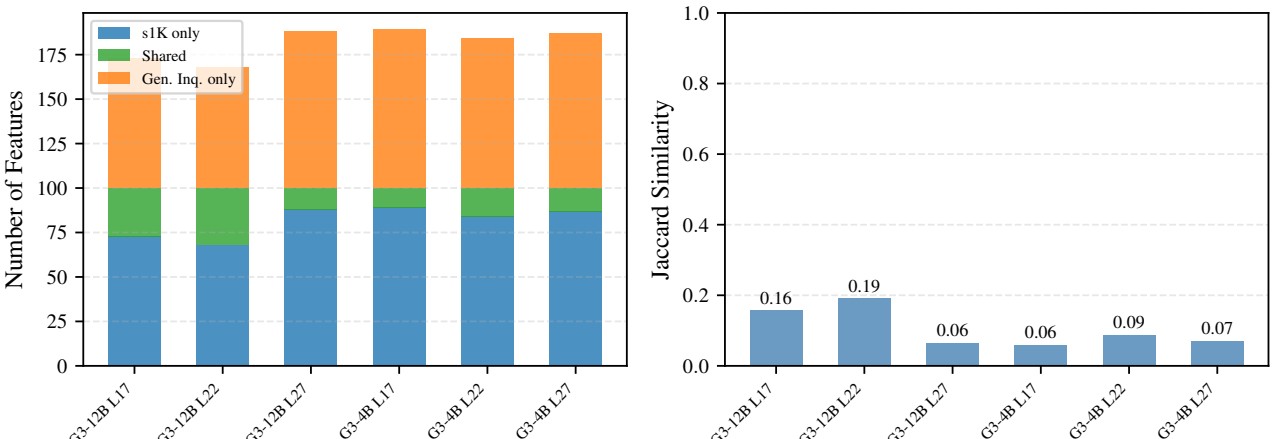

*Figure 10.* Feature overlap between s1K and General Inquiry CoT across models and layers.

# F. Hyperparameter settings

This section specifies all hyperparameters used throughout the experimental pipeline described in Sections 4 and 5. Unless otherwise stated, these settings are shared across all models, layers, and datasets.

## F.1. Feature detection

Candidate reasoning features were identified using 1,000 samples from each corpus, reasoning and non-reasoning, with each sample truncated to a maximum of 64 tokens. Feature activations were aggregated using the maximum activation across tokens. In the main experiments, features were ranked by Cohen's $d$, while appendix experiments used alternative ranking metrics as described in Appendix D.

We imposed three simultaneous selection thresholds: minimum Cohen's $d$ of 0.3, maximum Bonferroni-corrected $p$-value of 0.01 using the Mann-Whitney $U$ test (Mann & Whitney, 1947), and minimum ROC-AUC of 0.6. From the features satisfying all criteria, we selected the top 100 ranked by the primary metric.

For token dependency analysis, we extracted the top 30 tokens per feature based on mean activation, requiring a minimum of five occurrences. We additionally extracted the top 20 bigrams with at least three occurrences and the top 10 trigrams with at least two occurrences. Activation collection used a batch size of 16 for all models.

## F.2. Token injection

Token injection experiments were conducted on the top 100 features selected in the detection stage. For each feature, we used 500 non-reasoning samples for the baseline condition and 500 non-reasoning samples for the injected condition, with all samples chunked to 64 tokens. All processing used a batch size of 16.

We evaluated eight injection strategies: `prepend`, `intersperse`, `replace`, `inject_bigram`, `inject_trigram`, `bigram_before`, `trigram`, and `comma_list`. For simple token strategies, namely `prepend`, `intersperse`, `replace`, and `comma_list`, we injected three tokens selected from the top 10 tokens for the feature. For bigram-based strategies, we injected two bigrams selected from the top 20 bigrams. For trigram-based strategies, we injected one trigram selected from the top 10 trigrams.

Feature classification followed Cohen's effect size conventions. Token-driven features required $d \geq 0.8$ with $p < 0.01$. Partially token-driven features required $0.5 \leq d < 0.8$ with $p < 0.01$. Weakly token-driven features required $0.2 \leq d < 0.5$ with $p < 0.05$. Features with $d < 0.2$ or $p \geq 0.05$ were classified as context-dependent. For each feature, the injection strategy yielding the largest Cohen's $d$ was used for final classification.

## F.3. LLM-guided falsification

For features classified as context-dependent in the injection stage, we randomly sampled up to 20 features per configuration for LLM-guided falsification as described in Section 4.4. The protocol was run for a maximum of 10 iterations per feature, with early stopping once three valid false positives and three valid false negatives were identified.

The activation threshold for validation was set to 0.5 times the maximum activation observed on reasoning samples for the feature. A false positive was considered valid if its maximum activation exceeded this threshold, while a false negative was considered valid if its maximum activation was below 0.1 times the threshold. In each iteration, the LLM generated five candidate false positives and five candidate false negatives.

Temperature settings varied by phase. Hypothesis generation and final interpretation used temperature 0.3 to promote consistency, while counterexample generation used temperature 0.8 to encourage diversity.

## F.4. Steering experiments

Steering experiments were conducted exclusively on Gemma-3-12B-Instruct at layer 22, using the top three features ranked by Cohen's $d$ on the s1K dataset. We evaluated two steering strengths, $\gamma = 0.0$ for the baseline condition and $\gamma = 2.0$ for positive amplification. The steering intervention was scaled by the maximum feature activation observed in the reasoning corpus.

Text generation used a maximum of 16,384 new tokens, sampling temperature 0.6, and nucleus sampling with top-$p$ set

to 0.95. All prompts followed the model's chat template. Evaluation used one-shot chain-of-thought prompting. For AIME 2024, performance was measured by exact numerical answer matching, while for GPQA Diamond, performance was measured by multiple-choice accuracy.

### F.5. Model and SAE configuration

All Gemma-3 models used the GemmaScope-2 sparse autoencoder release with 16,384 features per layer and small $\ell_0$ regularization. DeepSeek-R1-Distill-Llama-8B used an SAE with 65,536 features trained on reasoning-focused datasets. All models were run in bfloat16 precision on a single NVIDIA A100 80GB GPU.

For additional models reported in Appendix C, we used analogous configurations. Llama-3.1-8B used SAEs from the Llama Scope release with 32K features per layer. Gemma-2-9B and Gemma-2-2B both used SAEs with 16,384 features per layer from the corresponding GemmaScope release. Experiments using alternative ranking metrics in Appendix D employed identical model and SAE configurations, differing only in the feature ranking criterion.

# G. LLM-guided feature interpretation results

This section provides complete documentation of all LLM-generated feature interpretations, including high-activation examples with token-level visualization, refined interpretations, and generated counterexamples, for one representative configuration (Gemma-3-4B-Instruct, layer 22, General Inquiry CoT). For each feature, we show three high-activation examples from the reasoning corpus with tokens colored by activation strength (darker blue indicates higher activation), followed by the LLM's interpretation and classification, and examples of false positives (non-reasoning text that activates the feature) and false negatives (paraphrases of high-activating reasoning text that does not activate the feature).

## Feature 163

*High-Activation Examples:*

*Example 1:* Then, I needed to describe how attention mechanisms address this by allowing the decoder to selectively attend to different parts of the image during caption generation. Initially, I need to examine the steps involved in the

*Example 2:* AI excels at pattern recognition and data processing but struggles with common-sense reasoning and abstract thinking. I realized this limitation would directly impact its ability to grasp the nuances of jokes. Next, I thought

*Example 3:* of these areas, I should provide a concise explanation of how NLP is used and its benefits. For example, for chatbots, I can explain how NLP enables them to understand and respond to human language.

*Interpretation:* This feature detects a specific syntactic construction used to express procedural necessity and goal-oriented sequencing, characterized by an initial infinitive of purpose ('To [verb]...') followed by sequence markers ('first', 'then') and modals of obligation ('must', 'need', 'should'). It does not distinguish between internal cognitive planning (reasoning) and external physical instructions (navigation/policy), activating on both provided the grammatical structure is present.

*Classification:* Confound (Confidence: HIGH)

*False Positives (Non-reasoning content that strongly activates the feature):*

1. To reach the ancient temple ruins, tourists first need to traverse the dense jungle path. Then, you will cross the river using the old bridge. To navigate the terrain safely, you must wear appropriate...

2. To cross the dangerous mountain pass safely, hikers first need to navigate the steep ridge clearly marked on the trail map. Then, you must traverse the glacier efficiently to avoid the risk of falling...

3. To enforce the privacy policy efficiently, the corporation first must identify the data sources clearly defined in the user agreement. The legal department should then review the consent forms thoroug...

*False Negatives (Semantic paraphrases of high-activation samples that fail to activate):*

1. Addressing improvements in image captioning via attention mechanisms starts by grasping the limits of older models lacking such features. Traditional systems compress images into fixed vectors, causin...

2. Initial thoughts focused on why AI finds humor difficult. Humor relies on complex human traits missing in current software. Thinking about key elements like surprise and context came before everything...

3. Grasping the request for practical Natural Language Processing examples is the starting point. Effective handling involves looking at NLP in business and daily tech. Analysis of application areas like...

## Feature 13118

*High-Activation Examples:*

*Example 1:* effectively, I'll first identify the core challenges. These generally revolve around latency, throughput, data freshness, resource management, and scalability/availability. Then, for each challenge, I'll outline

*Example 2:* multiplying 63 by 400, I can first multiply 63 by 4, and then multiply the result by 100. This strategy simplifies the calculation because multiplying by

*Example 3:* PM). This involves finding the difference between the minutes and the hours. There are 30 minutes from 0:30 AM to 11:00 AM, and then

*Interpretation:* This feature detects procedural segmentation and distinctness, specifically triggering on vocabulary that emphasizes handling components individually (e.g., "separately", "individually") within a step-by-step structure. It is not specific to cognitive reasoning but rather identifies the linguistic pattern of breaking a task—whether mental or physical—into discrete, sequential actions.

*Classification:* Confound (Confidence: HIGH)

*False Positives (Non-reasoning content that strongly activates the feature):*

1. To pack the fragile china securely, I need to wrap the plates separately. First, I'll cushion the cups individually with paper. The box itself is reinforced. The move entails a loading phase. Then, I'...

2. To style the hair perfectly, I need to curl the sections separately. First, I'll spray the strands individually for hold. The heat itself is moderate. The look entails a setting phase. Then, I'll brus...

3. To iron the dress shirts correctly, I need to press the collars separately. First, I'll spray the cuffs individually with starch. The fabric itself is cotton. I must use a high heat setting for the be...

*False Negatives (Semantic paraphrases of high-activation samples that fail to activate):*

1. Real-time query systems face hurdles like latency or throughput so we should look at those core issues plus data freshness. Spotting these challenges helps us find targeted solutions for keeping syste...

2. Calculating the time from ten-thirty to noon means counting the minutes up to eleven plus the full hour that follows. This strategy gives us the total duration by combining the two parts.

3. Large number multiplication gets simpler with patterns like powers of ten. We can also use shortcuts for numbers like eleven or five to speed things up. These tricks make the calculation much faster.

## Feature 1123

*High-Activation Examples:*

*Example 1:* First, I should consider the importance of consistency. A consistent UI helps users learn the system quickly and reduces cognitive load. This leads me to the next point: clear feedback. Users need to know

*Example 2:* Initially, I need to examine the provided numbers. Jane has 7 stickers, and she gives 4 away. So, the operation I need to perform is subtraction. That must mean I subtract

*Example 3:* answer this question effectively, I first needed to consider the current state of AI research and development. I thought about the areas where I've been seeing the most buzz and advancements lately. I immediately thought

*Interpretation:* This feature detects a specific first-person syntactic template characterized by the dense usage of process-oriented adverbs (specifically "effectively," "comprehensively," "thoroughly," "accurately") combined with sequence markers ("first," "next") and the pronoun "I". It activates on this lexical cluster regardless of whether the subject matter is cognitive planning or physical action (e.g., cleaning a microphone or stocking a break room).

*Classification:* Confound (Confidence: HIGH)

*False Positives (Non-reasoning content that strongly activates the feature):*

1. To serve the colony effectively, I first comprehensively filtered the water supply. Next, I thoroughly inspected the storage tanks. I accurately measured the levels today. Therefore, I approach the ma...

2. To assist the developers effectively, I first comprehensively organized the server racks. Next, I thoroughly labeled the cables. I accurately connected the power supply. My approach indicates that the...

3. To support the developers effectively, I first comprehensively stocked the break room with fresh coffee beans. Next, I thoroughly organized the tangled power cables beneath the shared workstations. I ...

*False Negatives (Semantic paraphrases of high-activation samples that fail to activate):*

1. Solving the sticker problem requires basic arithmetic. Jane starts with 7 and gives away 4. Subtracting the given amount from the total yields the final count of stickers remaining.

2. Current trends in technology highlight Generative AI as a rapidly advancing field. Recent developments have seen models producing high-quality images and text, marking a significant shift in the capab...

3. Healthcare benefits significantly from machine learning integration. Key applications include improving diagnostics, tailoring medicine to individuals, and accelerating drug discovery. These technolog...

## Feature 282

*High-Activation Examples:*

*Example 1:* I need to break down what factors might affect that performance. I can start with the most obvious one: the data the model is trained on. If the data is flawed, the model will learn

*Example 2:* My approach begins with identifying these stages, which typically include idea generation, drafting, editing, character development, and world-building (especially for genres like fantasy and science fiction). Initially, I need

*Example 3:* solving the numerical problem provided. Next, I need to address the specific problem: -8 +5 - (-3) -2. To tackle this effectively, I will apply the rules

*Interpretation:* This feature detects the specific syntactic template of first-person procedural planning, characterized by the combination of self-referential pronouns ("I", "My") with sequencing markers ("First", "Then") and strategic nouns/verbs ("approach", "need to", "identify"). It functions as a style detector for the 'Chain of Thought' format rather than a semantic detector of reasoning, as it activates on any content using this 'First, I need to...' structure (including simple descriptions) but fails to detect identical reasoning processes when written in the passive voice or third person.

*Classification:* Confound (Confidence: HIGH)

*False Positives (Non-reasoning content that strongly activates the feature):*

1. To describe this software feature, I need to clarify the structure. First, I need to define the user inputs. Then, I will provide an explanation of the background calculation. Next, I need to list the...

2. My approach to reviewing this novel involves a literary explanation. First, I need to identify the author's objectives. Then, I must consider the narrative inputs and themes. Next, I will analyze the ...

3. To critique this novel effectively, I need to formulate a structured approach. First, I need to consider the narrative explanation. Then, I must identify the thematic inputs provided by the author. Ne...

*False Negatives (Semantic paraphrases of high-activation samples that fail to activate):*

1. Defining 'accuracy' in the context of AI serves as the starting point. It essentially measures how well a model performs tasks like classification. Following the definition, an analysis of performance...

2. An analysis of the inquiry highlights the role of AI in creative writing. To provide a comprehensive answer, one must examine the various phases of the writing process. Identifying stages such as idea...

3. Answering the question comprehensively requires outlining key strategies for database optimization. Indexing is a critical tool for faster data retrieval. Describing the purpose of indexes and showing...

## Feature 14872

*High-Activation Examples:*

*Example 1:* First, I should consider the division itself. 17 divided by 2 is 8.5. However, since I can't give half a candy, I need to consider the whole

*Example 2:* last, I need to figure out how many times 1 eraser can be taken out of the total of 4 erasers. This sounds like a division problem. First, I needed to understand the

*Example 3:* friends. This is a division problem. I should divide the total number of bananas (4) by the number of friends (2). So, I'll calculate 4 ÷ 2.

*Interpretation:* This feature detects the syntactic pattern of first-person procedural planning and self-narration, specifically triggering on the combination of first-person pronouns ('I', 'me') with modals of necessity or intent ('need', 'should', 'will'). It identifies the linguistic style of an agent explicitly articulating their next steps, regardless of whether the content is abstract mathematical reasoning or concrete physical instructions.

*Classification:* Confound (Confidence: HIGH)

*False Positives (Non-reasoning content that strongly activates the feature):*

1. To assemble this bookshelf, I first need to determine which board is the bottom piece. I should look for the pre-drilled holes near the edge. I will align the side panel with the base. I need to figur...

2. First, I need to determine if the car tires are properly inflated. I should look for the recommended pressure on the door label. I need to figure out which tire looks low. I will simply attach the air...

3. First, I need to determine the correct time to set this alarm clock. I should check the time on my phone. I need to figure out which button changes the hour digit. I will simply hold down the 'set' bu...

*False Negatives (Semantic paraphrases of high-activation samples that fail to activate):*

1. Analyzing the task involves splitting 17 candies between 2 kids without breaking any. The situation calls for integer division. 17 over 2 makes 8, leaving 1 behind. This calculation shows that 8 is th...

2. Review the provided data points. Mike holds 4 erasers total. Usage is 1 per day. Determining the duration involves dividing the total quantity by the daily consumption rate. The math is clear: 4 divid...

3. The problem asks for an equal share of bananas. With 4 bananas and 2 buddies, division is the necessary operation. Splits happen evenly here. Dividing 4 by 2 results in 2. Each friend receives exactly...

## Feature 491

*High-Activation Examples:*

*Example 1:* the ethical concerns of AI, I need to consider the various aspects of AI development and deployment where ethical issues can arise. First, I should consider the data used to train AI models. If the training

*Example 2:* , I need to consider various aspects of data handling and AI implementation. Initially, I need to examine the stages where data privacy is most at risk, from data collection to model training and deployment. First

*Example 3:* problems. My approach begins with recognizing that multilingual support involves more than just translating text. It encompasses a comprehensive adaptation of the software to different languages, cultures, and regional requirements. Initially, I need

*Interpretation:* This feature detects the syntactic template and punctuation of explicit procedural planning preambles, specifically the 'To [goal], I need to [action]' structure often found in Chain-of-Thought responses. It tracks the structural markers (periods, commas, 'First', 'effectively') of this specific self-reflective planning style, activating equally strongly on complex topics and trivial/synthetic examples provided they utilize this specific rhetorical template.

*Classification:* Confound (Confidence: HIGH)

*False Positives (Non-reasoning content that strongly activates the feature):*

1. To address the topic of invisible ink effectively, I need to consider the transparency of the liquid. First, I need to recall the secret message. My approach begins with analyzing the blank paper comp...

2. To address the topic of arranging the bookshelf effectively, I need to consider the visual appeal of the covers. First, I should outline a color-coding system. My approach begins with sorting the book...

3. To answer the question of how to brew tea comprehensively, I need to consider the water temperature. First, I need to identify the type of tea leaves. My approach begins with boiling the water to the ...

*False Negatives (Semantic paraphrases of high-activation samples that fail to activate):*

1. Mere translation fails to achieve true multilingual support; the software's core architecture needs modification. The code itself must change to accommodate the structural differences between language...

2. Ethical risks in AI development are most prominent in the data collection phase, where biases in training sets can be learned and amplified by the system if not carefully managed.

3. Surrounding information allows AI systems to interpret user intent correctly by providing the necessary background for ambiguous queries.

## Feature 4510

*High-Activation Examples:*

*Example 1:* for implicit constraints and requirements. The mention of "limited resources and time" suggests that I need to focus on efficiency and cost-effectiveness. This eliminates approaches that are resource-intensive upfront, such as

*Example 2:* easier to track. Next, I need to consider other methods people use, like aligning numbers by place value and adding column by column, carrying over when necessary. I should also think about ways to simplify

*Example 3:* resulting in P = 2 * length + 2 * width, which can also be written as P = 2 * (length + width). Next, I needed to address the units of

*Interpretation:* This feature detects the infinitive marker "to" specifically when used in syntactic structures indicating purpose (e.g., "To [verb]...") or necessity (e.g., "need to..."). It is a grammatical feature that identifies goal-oriented procedural language, activating on step-by-step plans regardless of whether the content is complex reasoning, a recipe, or a simple narrative action.

*Classification:* Confound (Confidence: HIGH)

*False Positives (Non-reasoning content that strongly activates the feature):*

1. To bake the perfect chocolate cake, I need to preheat the oven to 350 degrees. First, I need to sift the dry ingredients into a large bowl. To ensure the batter is smooth, I need to beat the eggs one ...

2. First, I need to hide before the guards come back. To stay alive, I need to reach the ventilation shaft in the ceiling. I need to move quietly. To open the grate, I need to use the small knife in my p...

3. To assemble this bookshelf, I need to identify the screws listed in the manual. First, I need to lay out all the wooden panels on the floor. To connect the sides, I need to use the allen wrench provid...

*False Negatives (Semantic paraphrases of high-activation samples that fail to activate):*

1. Understanding the central question regarding knowledge base expansion under tight resources is vital. Prioritizing strategies that maximize impact while minimizing effort is key. Initially, examining ...

2. Answering effectively requires considering various approaches for adding large sums. Breaking figures into hundreds, tens, and ones seems optimal, facilitating tracking and mirroring standard learning...

3. Calculating a rectangle's perimeter starts by recalling the definition: total distance around a shape. Adding lengths of all four sides achieves this. Logic dictates summing two lengths and two widths...

## Feature 3810

*High-Activation Examples:*

*Example 1:* the positive direction. I also need to provide a clear algebraic representation of this principle, such as `a - (-b) = a + b`, and then give concrete examples to demonstrate the rule

*Example 2:* ?" and "What are the different types of AI?" Next, I should think about the practical applications and real-world impacts of AI, which prompts questions like "What are the current applications of AI

*Example 3:* , I need to understand the question which is asking for practical examples of Natural Language Processing (NLP). To tackle this effectively, I should consider various aspects of NLP usage in both everyday technology and business.

*Interpretation:* This feature detects a specific lexical and syntactic template used for procedural planning, characterized by the pattern 'First, I need to [verb]' or 'To tackle this... effectively'. It responds strictly to the combination of first-person modal necessity ('I need to') followed by analytical verbs ('identify', 'define', 'outline'), regardless of whether the subject matter is cognitive reasoning, sports strategy, or creative writing.

*Classification:* Confound (Confidence: HIGH)

*False Positives (Non-reasoning content that strongly activates the feature):*

1. To tackle this dream effectively, I need to identify the shifting shadows. First, I should define the nature of the fog surrounding me. I need to outline the path through the void and establish a conn...

2. To tackle this awkward dinner effectively, I need to identify a safe topic of conversation. First, I should define the mood of the room to avoid tension. I need to outline my anecdotes and establish a...

3. To tackle this sculpture effectively, I need to identify the natural grain of the wood. First, I should define the posture of the central figure. I need to outline the rough cuts and establish the cor...

*False Negatives (Semantic paraphrases of high-activation samples that fail to activate):*

1. The assignment is to demonstrate how to subtract negative integers. Step one is grasping the essence of subtraction and how it ties to addition. Subtraction acts like adding the inverse, so removing a...

2. The query seeks real-world samples of Natural Language Processing. A solid response involves exploring NLP's role in daily tech and business. I will start by breaking down the specific areas using thi...

3. First, we have to retrieve the main formula for a rectangle's area. The total space is the product of length and width. So, the rule is 'area = length * width'. To add depth, I should also talk about ...

## Feature 1148

*High-Activation Examples:*

*Example 1:* an AI model better, I need to break down the training process into key components. First, I need to consider the data itself – its quality, preparation, and how it's fed into the

*Example 2:* multilingual software application support, I first need to break down the overall problem into smaller sub-problems. My approach begins with recognizing that multilingual support involves more than just translating text. It encompasses a comprehensive adaptation

*Example 3:* To tackle this effectively, I should break down the answer into two main parts: the considerations and the types of services. For the considerations, I need to think about what someone would look for when

*Interpretation:* This feature detects a specific lexical cluster of formal vocabulary related to structural decomposition, organization, and thoroughness (e.g., "break down," "categorize," "thoroughly," "structured"). It activates on the presence of these specific terms whether they are used in a meta-cognitive planning context or in static descriptions of systems, manuals, and software updates. It fails to detect the semantic concept of problem decomposition when expressed in simpler or more casual language.

*Classification:* Confound (Confidence: HIGH)

*False Positives (Non-reasoning content that strongly activates the feature):*

1. Version 2.0 has been implemented to thoroughly address user feedback. We have categorized the settings menu to break down complex options into distinct panels. This update introduces a structured work...

2. The corporate safety manual is structured to thoroughly address workplace hazards. It categorizes the potential risks to break down the emergency protocols into distinct action plans, focusing on empl...

3. The corporate audit framework is employed to thoroughly address the question of financial transparency and compliance. It carefully categorizes the transaction logs to break down the complex revenue s...

*False Negatives (Semantic paraphrases of high-activation samples that fail to activate):*

1. Regarding the question on supporting software in multiple languages, I will split the big task into smaller bits. It starts with realizing that this involves more than just translating words. It requi...

2. Regarding the question on supporting software in multiple languages, I will split the big task into smaller bits. It starts with realizing that this is more than just translating text. It involves ful...

3. The user asks how to apply AI to personal tasks with few resources. Because that subject is huge, I will select particular spots where it works. I'll start with writing and art tools.

**Feature 6048**

*High-Activation Examples:*

*Example 1:* question comprehensively, I need to outline several key strategies for optimizing database query performance. First, I should consider indexing. Indexes are crucial for speeding up data retrieval, so I need to explain their purpose,

*Example 2:* question effectively, I need to explain what estimation is in the context of addition and then describe some common strategies used for estimation. First, I need to define estimation as a method for approximating numbers to simplify

*Example 3:* question effectively, I need to explain how subtraction can be understood in terms of adding negative numbers. First, I need to define the concept of an additive inverse (or negative) of a number. This

*Interpretation:* This feature specifically detects the first-person pronoun "I" when used in the context of meta-cognitive planning, self-narration, or outlining a response strategy (e.g., "I need to", "I should"). It is strictly tied to the first-person perspective and the syntactic structure of an agent explicitly stating their intentions, rather than the semantic content of reasoning itself.

*Classification:* Confound (Confidence: HIGH)

*False Positives (Non-reasoning content that strongly activates the feature):*

1. To answer this question about the course structure, I need to outline the core topics effectively. First, I should consider the introductory module. I need to identify the required textbooks and expla...

2. To answer this question about the perfect sourdough, I need to identify the core biological processes of fermentation. First, I should consider the activity of the wild yeast. I need to explain how te...

3. To answer this question effectively, I need to outline the core principles of interior design for small spaces. First, I should consider the layout and flow of the room. I need to identify how lightin...

*False Negatives (Semantic paraphrases of high-activation samples that fail to activate):*

1. To provide a comprehensive answer regarding database query performance, an outline of key strategies is essential. Indexing functions as a primary consideration. Because indexes speed up data retriev a...

2. Tackling this inquiry effectively involves defining estimation within the context of addition, followed by a description of common strategies. The definition should characterize estimation as a method...

3. Understanding subtraction through the lens of adding negative numbers requires a clear explanation. A crucial initial step involves defining the additive inverse. This foundation establishes that subt...

**Feature 791**

*High-Activation Examples:*

*Example 1:* First, I should consider the data itself. If AI systems are dealing with sensitive data, robust security measures must be in place, like encryption and anonymization. I need to highlight the risk

*Example 2:* additional marbles she finds, which is 2. To determine the total number of marbles Olivia has, I must combine these two quantities. This means I should add the number of initial marbles to the number

*Example 3:* To find the total number of seashells, I should add these two quantities together. So, I need to calculate 5 + 2. Performing the addition, 5 + 2 equals 7

*Interpretation:* This feature detects the specific syntactic sequence of a comma following an introductory transition (like "First" or "To answer...") immediately preceding the phrase "I need to" or "I must." It identifies a specific structural template often used in Chain-of-Thought prompting but activates on this syntax regardless of whether the semantic content is logical, nonsense, or creative.

*Classification:* Confound (Confidence: HIGH)

*False Positives (Non-reasoning content that strongly activates the feature):*

1. To answer this question comprehensively, I need to consider the existential dread of a melted snowman. First, I need to identify where the carrot nose went, as this is the most pressing mystery of the...

2. First, I need to identify the melody within the noise. To compose this symphony, I need to consider the rhythm of the city streets and how the traffic sounds blend into a chaotic harmony.

3. To answer this question comprehensively, I first need to consider the possibility that we are all just butterflies dreaming. I should identify the color of my wings and determine if I can fly towards ...

*False Negatives (Semantic paraphrases of high-activation samples that fail to activate):*

1. Determining the suitability of AI for sensitive tasks requires analyzing the specific aspects that define sensitivity. Usually this implies handling regulated or confidential data. Consequently the re...

2. The calculation starts with the 7 marbles Olivia owns originally. Next comes accounting for the 2 additional marbles she discovered later. Finding the total demands combining these two separate quanti...

3. Recalling the standard formula for a rectangle's area is the prerequisite step. Multiplying length by width yields the area. The core concept is 'area = length * width'. Making the answer complete als...

**Feature 444**

*High-Activation Examples:*

*Example 1:* to break down the process into manageable steps: converting to improper fractions, finding a common denominator, subtracting the fractions, converting back to a mixed number, and then addressing the borrowing scenario. First,

*Example 2:* perform the calculation carefully. My approach begins with setting up the subtraction: 100 - 27 — Initially, I need to examine the ones column. I can

*Example 3:* down the answer into key strategies. Let me start by analyzing what makes a conversation seamless and engaging. This leads me to consider defining clear goals, mapping user journeys, and structuring the flow hierarchically.

*Interpretation:* This feature functions as a semantic detector for specific abstract, categorical, or procedural nouns (e.g., "process," "problem," "sentence," "recipe," "topic") that label systems, tasks, or structural elements. While these words frequently appear during planning (hence the initial hypothesis), the feature activates reliably on these tokens in non-reasoning contexts such as technical manuals, formal narratives, and descriptive writing, indicating it is tied to the vocabulary rather than the cognitive process.

*Classification:* Confound (Confidence: HIGH)

*False Positives (Non-reasoning content that strongly activates the feature):*

1. The editor circled the **sentence** in red. She noted that the **structure** was awkward and the **topic** was unclear. The entire **paragraph** lacked coherence. She recommended rewriting the **text*...

2. The manual outlines the installation **process**. It describes the **functionality** of each **element**. The **system** handles the data **retrieval** automatically. Users can configure the **setting...

3. The moderator posed the **question** to the panel. The **topic** of discussion was renewable energy. The expert provided a detailed **answer**. She analyzed the **issue** from a scientific **perspecti...

*False Negatives (Semantic paraphrases of high-activation samples that fail to activate):*

1. I need to determine what makes a bot chat effective. I'll look at clarity and engagement. I'll discuss the requirements, starting with creating a smooth exchange for the user.

2. I have to subtract 27 from 100. Because of the zeros, I need to borrow. I will place 100 on top of 27. The ones place is zero, so I must borrow from the neighboring columns before subtracting.

3. I have to take 27 away from 100. Because of the zeros, I need to borrow. I will place 100 on top of 27. The ones place is zero, so I must borrow from the left.

**Feature 34**

*High-Activation Examples:*

*Example 1:* Expressions (CTEs) often helps. Using `EXPLAIN` to analyze the execution plan is crucial because it shows how the database intends to run the query and highlights inefficiencies like full table scans. Rew

*Example 2:* list.sort()` modifies the list in-place. Then, I must explain the crucial role of the `key` parameter, which allows specifying a function to extract the sorting key from each object.

*Example 3:* that could be the information surrounding a user's query. Initially, I need to examine what kinds of information might constitute context for an AI. I must then connect this to how the AI processes this

*Interpretation:* This feature detects determiners (primarily "the", "its", "their") when they appear within specific procedural or instructional sentence templates, typically characterized by introductory markers like "First," "To," or "//" followed by intent phrases like "I need to identify" or "address the." It identifies the syntactic structure of outlining a primary step or definition, regardless of whether the content is cognitive reasoning, code documentation, or physical description.

*Classification:* Confound (Confidence: HIGH)

*False Positives (Non-reasoning content that strongly activates the feature):*

1. // First, to initialize the module, identify the core dependency. // The entire script relies on this file. // To handle the input, the function parses the string. // The core logic is defined in the ...

2. First, I need to identify the bird on the branch. To see the colors, I use my binoculars. The entire flock takes flight. I need to capture the image. To focus the lens, I turn the dial. The core marki...

3. // First, to execute the script, identify the core function. // The entire process runs in the background. // To handle the output, the system writes to the log. // Its configuration is loaded from th...

*False Negatives (Semantic paraphrases of high-activation samples that fail to activate):*

1. Start by grasping what constitutes complex database queries. Typically, such requests involve multiple tables, intricate filtering, plus massive datasets. Handling these efficiently demands attention ...

2. Addressing how one sorts lists of Python objects using attributes requires providing a full guide on various techniques. First, introducing `sorted()` functions and `list.sort()` methods is best, noti...

3. Answering how context affects AI outputs requires considering what 'context' means for AI and usage patterns. My approach starts by thinking about general definitions of context—circumstances forming ...

**Feature 13167**

*High-Activation Examples:*

*Example 1:* ,I need to identify the core capabilities of AI. These generally include machine learning, natural language processing, computer vision, robotics, and expert systems. I need to define each of these capabilities clearly,

*Example 2:* ,I need to determine the core issue the question is addressing, which is how to improve the precision of search queries. To tackle this effectively, I need to break down the concept of search query precision

*Example 3:* ,I need to consider the core aspects of LLMs and their functionalities to identify potential limitations. To tackle this effectively, I'll start by thinking about what LLMs are good at, which is

*Interpretation:* This feature functions as a precise n-gram detector for the sequence "First, I need to", specifically activating on the token "I" within this structure. It captures the syntactic initiation of a first-person sequential action or plan, but it is agnostic to the semantic content, activating equally on abstract cognitive planning and mundane physical chores.

*Classification:* Confound (Confidence: HIGH)

*False Positives (Non-reasoning content that strongly activates the feature):*

1. First, I need to remove the core of the apple. It is hard and inedible, so I use a small paring knife to carefully cut it out. Once that is done, I can slice the rest of the fruit into even rings for ...

2. First, I need to identify my suitcase on the crowded luggage carousel. There are so many black bags that look exactly alike, circling around and around. I squint my eyes under the fluorescent lights, ...

3. First, I need to catch my breath. I ran all the way from the subway station to the office building because I overslept and was running late. My heart is pounding in my chest, and I have to lean agains...

*False Negatives (Semantic paraphrases of high-activation samples that fail to activate):*

1. To initiate the process, one must outline the fundamental functions of artificial intelligence. These elements typically comprise machine learning, natural language processing, and robotics. Defining ...

2. The starting point is determining the main issue: improving search query precision. To handle this, breaking down the concept of accuracy is required, along with finding methods that influence it. The...

3. Let us begin by evaluating the primary aspects of Large Language Models to find their constraints. An effective analysis starts by looking at their strengths, which are mainly text processing based on...

**Feature 928**

*High-Activation Examples:*

*Example 1:* should prioritize strategies that maximize impact while minimizing time and effort.Initially, I should examine the question for implicit constraints and requirements. The mention of "limited resources and time" suggests that I need to

*Example 2:* tackle this effectively, I should use an example to illustrate this principle. Initially, I need to examine a simple arithmetic expression that involves subtracting a negative number. Let's choose `5 - (-3

*Example 3:* to identify that the core problem is the unlike denominators. To tackle this, I need to explain the concept of a common denominator, specifically the least common multiple (LCM). I should illustrate how to find

*Interpretation:* This feature detects specific syntactic constructions related to first-person agency, intent, and necessity, characterized by the combination of first-person pronouns ('I', 'me', 'we') with modals ('should', 'can', 'will') and infinitives ('to'). It is not a reasoning detector; rather, it identifies the grammatical structure of a speaker stating what they are doing,

need to do, or will do, regardless of whether the context is solving a complex logic puzzle, making casual social plans, or describing physical actions.

*Classification:* Confound (Confidence: HIGH)

*False Positives (Non-reasoning content that strongly activates the feature):*

1. I'll come over to your place directly after work. You simply need to let me know the time. I can bring the drinks, and I should probably pick up ice too. This allows me to help out. We typically eat l...

2. To open the box, I need to pull the tab. This allows me to lift the lid. You can see the contents inside. I should handle it carefully. I'll place it on the table directly. I've checked the seal. It s...

3. I should have told you sooner. It leads me to feel guilty. I'll make it up to you, I promise. You can trust me on that. I simply forgot the date. I've been so busy lately. I need to apologize properly...

*False Negatives (Semantic paraphrases of high-activation samples that fail to activate):*

1. Understanding the core question regarding knowledge base growth under tight resources is key. This requires prioritizing high-impact strategies with minimal time or effort. Checking for implicit const...

2. Answering this requires explaining subtraction of negative numbers and the link with addition. Defining the core principle comes first: subtracting a negative equals adding a positive. Using an exampl...

3. Adding fractions with different denominators demands outlining the process. Identifying the unlike denominators is the main issue. Explaining the common denominator concept, specifically the least com...

**Feature 934**

*High-Activation Examples:*

*Example 1:* for me is to recall the multiplication table. I remember that 7 times 8 is a standard multiplication fact. If I didn't remember it directly, I could break it down. For example

*Example 2:* So, (2/5) * 150 gives me the number of apples sold. After calculating the apples sold, I need to subtract that number from the original 150

*Example 3:* ` by itself on one side of the equation. To do this, I need to undo the addition of 5. The opposite operation is subtraction, so I should subtract 5 from *both*

*Interpretation:* This feature detects procedural discourse markers and sequential transitions used to organize step-by-step processes. It activates strongly on temporal connectors (e.g., "Then", "Next", "Initially", "So") and syntactic structures that introduce a sequence of actions (e.g., "To [action], I first..."), appearing in both logical reasoning chains and mundane physical instructions (like painting or cleaning). It tracks the linguistic structure of a sequence rather than the semantic content of reasoning.

*Classification:* Confound (Confidence: HIGH)

*False Positives (Non-reasoning content that strongly activates the feature):*

1. To paint the room effectively, I first needed to prime the walls. Initially, the color looked too dark. Then, it dried to a lighter shade. Next, I applied the second coat. So, the finish was smooth. I...

2. To clean the garage effectively, I first needed to move the boxes. Initially, the dust made me sneeze. Then, I swept the floor. Next, I organized the tools. So, the space became usable again. Now, the...

3. To assemble the table effectively, I first needed to sort the screws. Initially, the parts looked similar. Then, I read the diagram. Next, I tightened the bolts. This led to a sturdy frame. So, the ta...

*False Negatives (Semantic paraphrases of high-activation samples that fail to activate):*

1. Multiplication is essentially repeated addition. Seven times eight translates to adding the number seven eight separate times. Memorizing the multiplication table offers a faster route. 7 multiplied b...

2. Figuring out the remaining apples involves calculating the sold amount. The market sales accounted for two-fifths of the 150 total apples. Multiplying 150 by 0.4 results in 60 apples sold. Deducting t...

3. Isolating `x` in the equation `3x + 5 = 14` is the primary objective. The addition of 5 necessitates subtraction as the inverse operation. Removing 5 from both sides preserves equality and simplifies ...

## Feature 4010

*High-Activation Examples:*

*Example 1:* , I need to consider the number of marbles Olivia initially possesses, which is 7. Then, I need to account for the additional marbles she finds, which is 2. To determine the total

*Example 2:* , I need to identify the core mathematical operation required to solve the problem. The question asks how many cookies each friend receives when 9 cookies are shared among 3 friends. This indicates a division problem

*Example 3:* , I need to figure out the total number of stamps Lily started with, which is 6. Then, I need to subtract the number of stamps she lost, which is 1. So,

*Interpretation:* This feature functions as a precise n-gram or phrase detector for the sequence "First, I need to", specifically activating on the pronoun "I" within this rigid structure. It is semantically blind, activating with identical strength ($\sim$677) on both complex cognitive planning and mundane physical descriptions (e.g., wiping tables), provided the specific introductory phrase is used.

*Classification:* Confound (Confidence: HIGH)

*False Positives (Non-reasoning content that strongly activates the feature):*

1. First, I need to unlock the front door. The key is right here in my pocket. I truly struggle with this old lock sometimes. We are finally home after a long trip. I just want to get inside and sit down...

2. First, I need to wipe the table. It is covered in crumbs from dinner. I sure hope we have enough paper towels left. I truly dislike a messy kitchen. We just finished eating. I need to clean this up qu...

3. First, I need to put on my heavy coat. It is freezing outside today. I truly hate the winter cold. I just want to stay warm while we walk. We are going to the shops down the street. I need to find my ...

*False Negatives (Semantic paraphrases of high-activation samples that fail to activate):*

1. Start by noting the 7 marbles Olivia has originally. Add the 2 extra marbles she found to that amount for the total. The sum of 7 and 2 is 9. This means she has 9 marbles in total.

2. To calculate the share for each friend divide the 9 cookies by 3 people. This division problem of 9 over 3 results in 3. Each friend receives 3 cookies.

3. Lily established a collection of 6 stamps initially. She lost 1 stamp so subtraction is the right step. Calculating 6 minus 1 leaves 5. The final number of stamps is 5.

## Feature 1430

*High-Activation Examples:*

*Example 1:* dive into the technical details. So I should provide options for both online courses and books. For online courses, I should consider platforms like Coursera, edX, fast.ai, and Udacity

*Example 2:* subtract 5 from *both* sides of the equation. This maintains the balance of the equation. When I subtract 5 from both sides, I get `3x + 5 - 5

*Example 3:* extraction, and content recommendation. For each of these areas, I should provide a concise explanation of how NLP is used and its benefits. For example, for chatbots, I can explain how NLP enables them

*Interpretation:* This feature detects a specific cluster of technical, geometric, and categorical vocabulary (e.g., "cube", "rectangle", "category", "samples") alongside adverbs of precision (e.g., "accurately", "effectively", "thoroughly"). While these words frequently appear during the decomposition phase of formal problem-solving, the feature is lexical rather than functional, activating equally strongly on static descriptions of physical objects, data layouts, or artwork that utilize this specific vocabulary.

*Classification:* Confound (Confidence: HIGH)

*False Positives (Non-reasoning content that strongly activates the feature):*

1. The sculpture consists of a large **cube** resting on a wide **rectangle**. To view the **samples** **accurately**, observers should stand to the **left**. This angle reveals the **biased** perspectiv...

2. The textbook page illustrates the concept of numbers by showing a grid of **integers** and **fractions** inside a yellow **rectangle**. The accompanying text answers the student's **question** **compr...

3. The laboratory rack holds the test **samples** sorted by **category**. To record the **data** **accurately**, the technician scans the barcode on the **left**. The protocol manual outlines the storage...

*False Negatives (Semantic paraphrases of high-activation samples that fail to activate):*

1. We need a dozen even wedges from one pie. The simplest path is splitting the circle in half, then into quarters. One cut makes two bits, and a cross cut yields four.

2. The request seeks practical uses for language processing tech. A good response covers its role in both daily gadgets and corporate tools. We can begin by spotting main fields like automated chat agent...

3. The core task is adding speech control to a Python app. The work divides into steps: grabbing audio, changing sound to text, parsing commands, running actions, and signaling the person that it is fini...

## Feature 1171

*High-Activation Examples:*

*Example 1:* a number line. First, I need to consider what subtraction means visually on a number line — it means moving to the left. The first number in the subtraction problem is where I should start on the

*Example 2:* 2, dividing by 2 and then dividing by 2 again will give the same result as dividing by 4. For example, if I want to divide 20 by 4, I

*Example 3:* missing addend. First, I need to identify the basic components of an addition equation: the addends and the sum is the total you get when you add the addends together.

*Interpretation:* This feature detects a specific lexical cluster involving modals of necessity, ability, and intent (e.g., "need", "can", "wants") combined with conditional or resultative connectors (e.g., "if", "gives", "until", "where"). Rather than detecting the semantic process of reasoning, it identifies sentences that describe requirements and their subsequent outcomes or conditions, which appears frequently in 'step-by-step' planning but equally in recipes, sports commentary, and troubleshooting guides.

*Classification:* Confound (Confidence: HIGH)

*False Positives (Non-reasoning content that strongly activates the feature):*

1. This is a classic recipe where patience gives the best flavor. The dough needs to rest in a warm bowl until it doubles in size. If you can wait, the high heat gives it a perfect golden crust. The bake...

2. Here is the answer to the connection issue everyone is asking about. The system wants to update automatically. If you can click the prompt, it gives you immediate access. We need to address the server...

3. He wants the victory more than anything. If he gives his best effort on the field, he can succeed. The fans are asking for a final goal. He needs to run hard until the whistle blows. This match gives ...

*False Negatives (Semantic paraphrases of high-activation samples that fail to activate):*

1. Division works by subtracting the divisor repeatedly. Remove the specific amount from the starting total over and over. The cycle ends at zero. Counting the removals reveals the final answer.

2. Like terms are defined by sharing the same variable and power. In the expression 3x + 4y - 2x + 5y, certain elements align. Note that 3x and -2x both use x to the first power. Thus, they belong togeth...

3. Division functions fundamentally as repeated subtraction. One subtracts the divisor from the main number repeatedly. The process concludes once the value hits zero. The total count of these subtractio...

## Feature 14138

*High-Activation Examples:*

*Example 1:* natural language. First, I need to think about the initial stage, which is obtaining the text data itself. So, the first step would naturally be **Text Acquisition**. After acquiring the raw text,

*Example 2:* he's sharing them with 3 siblings. It's crucial to understand that Alex is sharing with *his* siblings, and not including himself in the distribution. To figure out how many toys

*Example 3:* define "borrowing" in the context of subtraction, highlighting that it's more accurately described as regrouping. My approach begins with identifying the core issue – when a digit in the subtrahend

*Interpretation:* This feature detects a specific lexical cluster of words commonly found in educational word problems, technical descriptions, and illustrative examples. It activates on a fixed set of nouns (e.g., "birds", "eggs", "interface", "tax", "students") and verbs (e.g., "sharing", "handle", "indicates", "constitutes") regardless of the surrounding context. While these words frequently appear in the setup phase of reasoning tasks (e.g., a math problem about sharing), the feature responds to the vocabulary itself rather than the syntactic structure or reasoning process.

*Classification:* Confound (Confidence: HIGH)

*False Positives (Non-reasoning content that strongly activates the feature):*

1. The main ingredient in the recipe is organic **eggs**. **She** is **sharing** the chocolate **cakes** **with** the family. The process involves **handling** the batter carefully. The taste **is** **tr...

2. The garden is full of wild **birds** and **flies**. The stone path is described **as** **geometric**. The broken shell **indicates** the presence of **eggs**. The environment **constitutes** a safe ha...

3. The new computer features a **robust** **interface**. The software is **optimizing** the processing speed. The guide is **about** how to **handle** errors. The screen **indicates** the system status.

*False Negatives (Semantic paraphrases of high-activation samples that fail to activate):*

1. To solve this efficiently, I must review the standard NLP workflow. My process starts by gathering raw data. We label this step Text Acquisition. Once I have the material, I usually find it disorganiz...

2. I need to extract the key figures from the problem statement. Alex possesses 9 toys. He allocates these items to 3 siblings. We must note that Alex gives them away rather than keeping any. To find the...

3. To solve this, I will review the standard NLP sequence. Collecting data happens first. We label this phase Text Acquisition. Raw inputs often contain noise and need cleaning.

# H. Dataset details

This section provides detailed descriptions of the datasets used to construct reasoning and non-reasoning corpora in our experiments (Section 5). We use two reasoning datasets with explicit chain-of-thought traces and one general-text dataset as a non-reasoning baseline.

## H.1. s1K-1.1

The s1K-1.1 dataset is a curated collection of 1,000 challenging mathematics problems, derived from the original s1K dataset introduced by Muennighoff et al. (2025). The questions span a range of mathematical domains, including algebra, geometry, number theory, and combinatorics, and are designed to require multi-step reasoning. In contrast to the original s1K dataset, which contains reasoning traces generated by Gemini Thinking, s1K-1.1 augments the same set of 1,000 questions with additional chain-of-thought traces generated by DeepSeek-R1, resulting in a mixture of Gemini- and DeepSeek-generated solutions. We use both types of traces in our experiments.

Each example consists of a problem statement and an associated reasoning trace that decomposes the solution into explicit intermediate steps. The traces are written in natural language and include algebraic manipulations, case analyses, and explanatory commentary. We treat the reasoning trace as the reasoning text and exclude the final answer when constructing inputs for feature analysis, in order to focus on intermediate reasoning behavior rather than answer tokens. For all experiments, inputs are truncated to a maximum of 64 tokens.

## H.2. General Inquiry Thinking Chain-of-Thought

To evaluate reasoning features beyond mathematics, we use the General Inquiry Thinking Chain-of-Thought dataset (Wensey, 2025). This dataset contains approximately 6,000 question-answer pairs with explicit chain-of-thought annotations. Unlike mathematics-focused datasets, General Inquiry spans a broad range of domains, including scientific reasoning, logical puzzles, everyday decision making, and philosophical or conceptual questions.

Each example includes a user question and a step-by-step reasoning trace that justifies the final answer. The diversity of topics and styles reduces the risk that results are driven by domain-specific vocabulary or formatting conventions. As with s1K-1.1, we use only the chain-of-thought portion of each example and truncate inputs to 64 tokens. For each experimental configuration, we randomly sample 1,000 examples from the dataset to form the reasoning corpus.

## H.3. Pile Uncopyrighted

As a source of non-reasoning text, we use the uncopyrighted subset of the Pile (Gao et al., 2020). The Pile is a large-scale English text corpus constructed from 22 heterogeneous sources, including academic writing, books, reference material, and web text. The uncopyrighted version removes all copyrighted components to support responsible use.

We randomly sample 1,000 text passages from this corpus and truncate each passage to 64 tokens. These samples are treated as non-reasoning text and are used as the baseline distribution in contrastive feature detection (Section 4.2) and token injection experiments (Section 4.3). While some passages may contain implicit reasoning, the dataset does not include explicit chain-of-thought traces or structured problem-solving discourse, making it a suitable non-reasoning comparison corpus.

# I. Benchmark details

This section describes the external benchmarks used to evaluate the behavioral effects of feature steering (Section 5.5). These benchmarks are used only as supplementary evaluations and do not form the basis of our primary conclusions.

## I.1. AIME 2024

We evaluate mathematical reasoning performance using problems from the American Invitational Mathematics Examination (AIME) 2024 (Zhang & Math-AI Team, 2024). AIME is a well-established mathematics competition at the advanced high school level, consisting of problems that require multi-step symbolic reasoning and careful algebraic manipulation. The 2024 edition includes 30 problems drawn from the AIME I and AIME II contests.

Each problem has a single integer answer, and detailed official solutions are available. In our experiments, we use the problem statements as prompts and evaluate model outputs based on exact match with the ground-truth numerical answers. We use one-shot chain-of-thought prompting and allow long generation lengths to avoid truncating reasoning traces.

## I.2. GPQA Diamond

We also evaluate scientific reasoning using the Diamond split of the Graduate-Level Google-Proof Question Answering benchmark (GPQA) (Rein et al., 2024). GPQA is a multiple-choice question answering dataset constructed by domain experts in biology, physics, and chemistry. The questions are intentionally designed to be difficult for both non-experts and state-of-the-art language models, even with access to external resources.

The Diamond split consists of the most challenging subset of questions and is commonly used as a stress test for advanced reasoning capabilities. Expert annotators achieve substantially higher accuracy within their own domains than non-expert validators, highlighting the depth of domain knowledge required. Due to the sensitivity of the dataset and the risk of leakage, we do not reproduce or paraphrase individual questions in this paper.

In our steering experiments, we evaluate accuracy on the multiple-choice answers using one-shot chain-of-thought prompting. As with AIME 2024, these results are reported as supplementary evidence and are interpreted cautiously, since performance changes alone do not imply that a steered feature encodes a genuine reasoning mechanism.

## J. Licenses and responsible use

We carefully adhere to the licenses and usage terms of all datasets, models, and benchmarks used in this study. For non-reasoning text, we use the uncopyrighted subset of the Pile dataset. The s1K-1.1 dataset and the General Inquiry Thinking Chain-of-Thought dataset are released under the MIT license.

We evaluate open-weight language models released under permissive or research-friendly terms. The Gemma 2 and Gemma 3 models are used in accordance with the Gemma Terms of Use. The Llama 3.1 models are used under the Llama 3.1 Community License Agreement. The DeepSeek distilled models are released under the MIT license. Sparse autoencoders from the Gemma Scope and Gemma Scope 2 releases are licensed under the Creative Commons Attribution 4.0 License, and the Llama Scope release is licensed under the Apache License 2.0.

For evaluation benchmarks, AIME 2024 is used under the Apache License 2.0, and GPQA is released under the Creative Commons Attribution 4.0 License.

Figure 1 incorporates icons created by karyative and Bartama Graphic from the Noun Project, licensed under the Creative Commons Attribution 3.0 License (CC BY 3.0). The icons have been resized and stylistically adapted for consistency with the figure design. Use of these icons complies with the license requirements, including attribution to the original creators and indication of modifications.

All experiments are conducted for research purposes only. We do not release any new model weights or datasets, and we do not modify or redistribute licensed resources beyond what is permitted by their respective terms. We encourage future work building on this study to similarly respect licensing constraints and responsible research practices.

