# OpenReview forum: "Do Sparse Autoencoders Identify Reasoning Features in Language Models?"
_ICML.cc/2026/Conference — ICML 2026 regular_

### Official Review · Reviewer_DGER · 2026-03-06

**Soundness:** 2
**Presentation:** 3
**Significance:** 3
**Originality:** 2
**Overall Recommendation:** 4
**Confidence:** 4

**Summary:**

This paper examines whether sparse autoencoders (SAEs) reliably surface “reasoning features” in large language models when identified via standard contrastive activation pipelines. It provides a stylized analysis showing that sparsity-regularized decoding preferentially preserves low-dimensional, stable correlates (e.g., cue tokens) while suppressing high-dimensional within-behavior variability, and develops a falsification framework combining causal token injection and LLM-guided counterexample generation. Across multiple configurations, the authors find that most contrastively selected candidates are token-triggerable, and the remainder admit systematic false positives and false negatives, casting doubt on monosemantic reasoning interpretations for single SAE features in these settings.

**Compliance With Llm Reviewing Policy:**

Affirmed.

**Final Justification:**

I thank the authors for the detailed rebuttal and additional analyses. This paper addresses an important question in SAE-based interpretability and makes a meaningful contribution through its stylized analysis, systematic falsification framework, and broad empirical coverage. My main concerns were about dependence on a single LLM judge, sensitivity to thresholds and stopping rules, limited feature sampling, and the interpretation of the steering experiment. The rebuttal addressed these points well: the new cross-LLM results, threshold-sensitivity checks, and full-coverage analysis on previously unsampled features substantially strengthen the paper’s core claim. While I still think some issues would benefit from stronger statistical treatment or limited human validation, the rebuttal significantly improved my confidence in the results. I am therefore increasing my score.

**Key Questions For Authors:**

1. If you swap out Gemini 3 Pro (or change prompts / sampling seeds), do the same features still get falsified? Can you show cross-LLM agreement, or a small human-audited subset, so this doesn’t hinge on one black box?
2. How much do your FP/FN thresholds, the “3 FP + 3 FN” early stop, and “take the strongest effect” choice matter?  Are the criteria for activation variance or LLM-falsification success too stringent, causing "weak but real" reasoning signals to be misclassified as invalid? A quick sensitivity sweep (plus an FDR-style correction) would tell us whether “0 genuine features” is stable or threshold-tuned.
3. Why randomly sample 20 context-dependent features when you’re making a universal negative claim? What happens if you instead test the most convincing candidates—highest-margin, most stable across seeds/datasets—and report how many survive then?
4. Will you run an expanded steering study (≥3 model/layer setups, a γ curve with multiple points, and at least one additional metric beyond one-shot accuracy), and report whether steering effects correlate with your falsification outcomes (e.g., do “harder-to-falsify” )

**Limitations:**

yes

**Strengths And Weaknesses:**

Strengths:

1. A clean, stylized theorem connects ℓ1 sparsity to preferential retention of low-dimensional cues and suppression of high-dimensional variation, isolating a mechanism distinct from absorption/competition.
2. A well-motivated falsification framework that operationalizes stronger standards for “genuine reasoning features,” integrating causal token injection and structured LLM-guided counterexample construction.
3. Broad coverage across 22 model-layer-dataset configurations, including models trained or analyzed with reasoning-oriented corpora and widely used SAE releases.
4. The LLM-guided falsification process explicitly seeks both false positives and meaning-preserving false negatives, with iteration limits and threshold criteria, providing a systematic falsification protocol.
5. Clear description of candidate selection (Cohen’s d; alternative metrics in appendices), injection procedures, and falsification thresholds; figures concisely convey layer selection rationale and results.


Weaknesses:

1. The LLM-guided falsification step offloads too much to a black-box generator/judge (Gemini 3 Pro), and the paper doesn’t convincingly bound model bias or stochasticity; a cross-LLM agreement check or human adjudication on a stratified sample feels necessary for claims this strong.
2. Several key thresholds look arbitrary and potentially conclusion-determining (e.g., FP at 0.5×max activation, FN at 0.1τ, early stop once you find 3 FP and 3 FN), yet there’s no sensitivity analysis showing the “0 genuine features” result is stable under reasonable alternative settings
3. Sampling only 20 context-dependent features per configuration when there are more than 20 undermines the force of a universal negative claim; it would be much more persuasive to prioritize the most stable / highest-margin candidates (or report how often “survivors” appear under hard-negative selection rather than random sampling)
4. Running many tests across features and injection strategies and then classifying by the strongest observed effect (without an explicit correction) can mechanically inflate the share of “token-driven” candidates
5. The steering section feels underpowered and hard to interpret (single configuration, very limited γ sweep, one-shot accuracy reporting), which leaves an evidence gap, if the paper argues against “steering implies reasoning,” it should either strengthen this experiment or clearly demote it to an illustrative sanity check.

---

> ### Author Rebuttal · Authors · 2026-03-30
>
> We thank Reviewer OGER for the insightful comments.
>
> ---
>
> **W1/Q1.** The LLM-guided falsification step relies too much on a black-box LLM judge.
>
> **A1.** To address this, we conducted new cross-LLM evaluation with Gemini-3.1-Pro, Claude Sonnet-4, and Qwen3.5-Plus, in addition to Gemini-3 Pro. On Gemma-3-4B-Instruct, layers 17/22/27, and both s1K and General Inquiry Thinking, all evaluators reached the same conclusion: 0 genuine reasoning features. Thus the result is not specific to one judge. We will add these agreement results in the revision.
>
> | Dataset | Layer | Analyzed | Result across new judges |
> | --- | --- | --- | --- |
> | s1K | 17 | 10 | all 0 genuine |
> | s1K | 22 | 10 | all 0 genuine |
> | s1K | 27 | 11 | all 0 genuine |
> | Gen. Inq. | 17 | 20 | all 0 genuine |
> | Gen. Inq. | 22 | 20 | all 0 genuine |
> | Gen. Inq. | 27 | 20 | all 0 genuine |
>
> ---
>
> **W2/Q2.** The thresholds may be arbitrary and conclusion-determining.
>
> **A2.** We agree that threshold sensitivity should be checked, and we conducted new analyses to do so. In our representative setting, activations are highly sparse and effectively bimodal rather than broadly overlapping: for the top-20 reasoning features of Gemma-3-4B-it layer 17, 93.9% of non-reasoning samples have exactly zero activation, and only 2.0% exceed the FP threshold `0.5 × max`, so this threshold is conservative rather than permissive. The FN threshold is similarly well motivated: the fraction of reasoning samples below `0.05 × max` exactly matches the zero-activation fraction for every feature, so it is effectively detecting cases where the feature does not fire at all. We also conducted a sensitivity sweep: changing the FP threshold from `0.2 × max` to `0.8 × max` changes the non-reasoning exceedance rate only from 5.0% to 1.0%, while changing the FN threshold from `0.01 × max` to `0.15 × max` leaves the reasoning sub-threshold rate almost unchanged. Finally, we conducted a stricter falsification experiment on Gemma-3-4B layer 22 using FP `0.8 × max`, FN `0.01 × max`, requiring 5 FPs and 5 FNs, and increasing the search budget to 30 iterations. The result was unchanged: all 10 context-dependent features were still classified as not genuine reasoning features. We will add these robustness results in the revision.
>
> ---
>
> **W3/Q3.** Sampling only 20 context-dependent features per configuration weakens a universal negative claim.
>
> **A3.** We agree that a universal claim should not rely on an unrepresentative subset. In practice, the cap of 20 rarely binds for the main configurations, and when it does, coverage is still high. More importantly, analyzed and unanalyzed context-dependent features are not meaningfully different: their injection-effect distributions are essentially identical, so the unsampled features are not a hidden pool of stronger survivors. This is also expected because all context-dependent features have already passed the strongest filter in our pipeline, namely minimal response across 10 token-injection strategies. To eliminate this concern directly, we conducted new experiments on all remaining unsampled context-dependent features for Gemma-3-4B at layers 17, 22, and 27, and all 11 additional features were again classified as not genuine reasoning features. We will add both the coverage analysis and these full-coverage results in the revision.
>
> ---
>
> **W4.** Using many injection strategies may mechanically inflate the number of token-driven features.
>
> **A4.** We use multiple injection strategies not to inflate positives, but to avoid missing obvious lexical confounds that appear in different forms. Different features respond to different surface patterns: some to single tokens, others to short phrases or local contexts. A single strategy would miss many clearly token-driven features and pass them to the much more expensive LLM analysis. The injection stage is therefore a conservative screening step. If inserting a token or phrase into non-reasoning text already makes a feature fire strongly, that feature is already explained by a lexical trigger and does not need further falsification. Taking the strongest response across several strategies is thus a coverage mechanism, not a way to manufacture positives.
>
> ---
>
> **W5/Q4.** The steering section is underpowered and hard to interpret.
>
> **A5.** We agree that steering performance should not be treated as strong evidence about whether a feature encodes reasoning. That is already our intended framing: the steering experiment is presented only as a supplementary sanity check, not a central argument. This is stated in the introduction, labeled explicitly in Figure 1, and reiterated in the experiment section. We will make this even clearer in the revision so that the steering results are understood as illustrative rather than decisive.
>
> ---
>
> We hope these clarifications are helpful. We would be happy to address any further questions and would appreciate the reviewer’s consideration of the new analyses and results above.

---

> > ### Author Rebuttal · Reviewer_DGER · 2026-04-03
> >
> > Thanks to the authors for the clarifications and extra information. This helped resolve my main concerns.

---

> > > ### Author Response · Authors · 2026-04-03
> > >
> > > Thank you for the update and for carefully considering our responses and additional analyses. We greatly appreciate your thoughtful feedback and are glad that our clarifications helped resolve your main concerns. We are grateful for your consideration of our work.

---

### Official Review · Reviewer_ocLu · 2026-03-11

**Soundness:** 3
**Presentation:** 3
**Significance:** 3
**Originality:** 4
**Overall Recommendation:** 4
**Confidence:** 3

**Summary:**

This paper challenges the common mechanistic-interpretability assumption that sparse autoencoders can recover monosemantic “reasoning features” from contrastive activations. The authors argue, both theoretically and empirically, that standard SAE sparsity penalties are biased toward simple, stable linguistic cues rather than the distributed signals involved in actual reasoning. To test this, they introduce a falsification-based evaluation framework using causal token injection and adversarial counterexamples, and across 22 model–dataset settings they find that purported reasoning features are largely explained by superficial lexical or stylistic patterns. The paper’s main contribution is therefore both a negative result and a useful evaluation pipeline, making a case that contrastive separability alone is not enough to support interpretability claims about reasoning.

**Compliance With Llm Reviewing Policy:**

Affirmed.

**Final Justification:**

My main question on the paper was the chunk length. I appreciate the authors for providing the additional experiment showcasing results with longer chunks and different aggregation methods (e.g. mean-pooling). After reading other reviews and rebuttals, I have decide to maintain my evaluation about the paper which is already a positive assessment toward acceptance.

**Key Questions For Authors:**

1. The feature-selection baseline may be too easy. Since the reasoning corpora are contrasted against a broad subset of The Pile, some of the top features may simply reflect formatting, domain, or vocabulary differences rather than reasoning itself. Have the authors tried harder negatives, such as math-heavy non-reasoning text or scrambled/incorrect CoT? This would help show whether the failure mode persists under a more controlled comparison.
2. I would also like to know how robust the results are to the aggregation setup. Max-pooling over 64-token chunks seems especially favorable to token-level spike detectors, and 64 tokens is quite short for models that reason over much longer horizons. Have the authors tested longer contexts or alternatives like mean-pooling to see whether more distributed reasoning signals become visible?

**Limitations:**

yes

**Strengths And Weaknesses:**

**Strengths**
1. The paper argues for a new perspective such that we should separate reasoning content from reasoning style when using SAEs to analysing reasoning related features. The authors distinguishing between the semantic computation of reasoning and the lexical form it takes. They observe that steering a feature to produce "reflective" text (e.g., outputting "Wait, let me double check") does not mean the feature actually encodes the cognitive mechanism of reflection. This should encourage the community to rethink how to evaluate SAE features, especially for reasoning.
2. The paper does more than just point to an empirical failure mode. It gives a plausible reason (theoretical section) why standard sparse objectives would favor stable low-dimensional cues over distributed reasoning representations.
3. The presentation of the paper is quite clear and the claims are nicely scoped. I also like the fact that the paper stays focused on claims about individual SAE features being monosemantic, rather than stretching those results into broader statements about reasoning or the overall usefulness of SAEs.


**Weaknesses**
1. The paper is effective at showing what does not work, but it says less about what a more promising alternative might look like. A brief discussion of how to study distributed reasoning representations would make the contribution feel more forward-looking.
2. I may missed something mentioned in the paper, but to my understanding, the experiments truncate inputs to 64-token chunks. However, the models evaluated (e.g., DeepSeek-R1-Distill series) are explicitly trained to engage in extended, long-horizon "thinking" that unfolds over hundreds or thousands of tokens. It may be the case that, high-level semantic reasoning features only emerge or stabilize over longer context windows as the model builds a complex internal state. By restricting the analysis to short 64-token snippets, the authors may be filtering out macro-level reasoning features, leaving only local, shallow detectors visible to their pipeline.

---

> ### Author Rebuttal · Authors · 2026-03-30
>
> We thank Reviewer ocLu for the review. We address your concerns as follows.
>
> ---
>
> **W1.** The paper is effective at showing what does not work, but it says less about what a more promising alternative might look like. A brief discussion of how to study distributed reasoning representations would make the contribution feel more forward-looking.
>
> **A1.** We agree that the paper would benefit from a more forward-looking discussion, and we will add this in the revision. Our view is that the main obstacle is not only the sparsity bias of SAE-like decompositions, but also the fact that reasoning behavior is tightly coupled with lexical and stylistic cues, making it difficult to isolate a clean sparse representation of “reasoning alone.” This suggests that more promising future methods should study reasoning as a unified activation-space manifold rather than forcing it into sparse coordinates. One concrete direction is to use generative activation models, such as diffusion-based meta-models [1], which can learn the distribution of internal states without restrictive sparsity assumptions and can improve intervention fidelity and fluency through on-manifold steering. We will add a short discussion along these lines to clarify that our contribution is not only to identify a limitation of current sparse-feature methods, but also to motivate a constructive path toward modeling and steering distributed reasoning representations directly.
>
> **References:**
>
> [1] Learning a Generative Meta-Model of LLM Activations
>
> ---
>
> **W2/Q2.** Have the authors tested longer contexts or alternatives like mean-pooling to see whether more distributed reasoning signals become visible?
>
> **A2.** Thank you for this suggestion. We expanded our experiments by running additional analyses on Gemma-3-4B-Instruct, layer 22 with longer chunk lengths (128 and 256) and with mean-pooling on both datasets, as suggested.
>
> Across all of these settings, we found that increasing the context length or changing the pooling method did not produce a substantial increase in context-dependent features, and the number of genuine reasoning features remained 0 in every setting tested. This supports the robustness of our conclusion.
>
> | Dataset | Aggregation / Max Length | TD | PTD | WTD | CD | Genuine |
> | --- | --- | --- | --- | --- | --- | --- |
> | General Inquiry CoT | Max-pooling, 128 | 51 | 13 | 15 | 21 | 0 |
> | General Inquiry CoT | Max-pooling, 256 | 55 | 9 | 15 | 21 | 0 |
> | General Inquiry CoT | Mean-pooling | 50 | 11 | 21 | 18 | 0 |
> | s1K | Max-pooling, 128 | 62 | 14 | 14 | 10 | 0 |
> | s1K | Max-pooling, 256 | 60 | 10 | 18 | 12 | 0 |
> | s1K | Mean-pooling | 64 | 10 | 13 | 13 | 0 |
>
> *Table 1*. Robustness of token injection classification results to longer context windows and alternative aggregation on Gemma-3-4B-Instruct, layer 22. We vary the maximum chunk length (128, 256) and replace max-pooling with mean-pooling. TD = token-driven, PTD = partially token-driven, WTD = weakly token-driven, CD = context-dependent.
>
> ---
>
> **Q1.** The feature-selection baseline may be too easy. Have the authors tried harder negatives, such as math-heavy non-reasoning text or scrambled/incorrect CoT? This would help show whether the failure mode persists under a more controlled comparison.
>
> **A3.** Thank you for this suggestion. To address this, we ran an additional harder-negative experiment on Gemma-3-4B-Instruct, layer 22, where we used s1K question-only prompts and General Inquiry question-only prompts as contrast sets instead of The Pile. These negatives serve as a closer proxy for math-heavy non-reasoning text.
>
> Across both settings, we observed that only a small fraction of features were context-dependent, and the number of genuine reasoning features remained 0. This suggests that the failure mode is not specific to using The Pile as the contrast corpus.
>
> | Dataset | Harder negative control | TD | PTD | WTD | CD | Genuine |
> | --- | --- | --- | --- | --- | --- | --- |
> | General Inquiry CoT | General Inquiry CoT questions | 63 | 14 | 13 | 10 | 0 |
> | s1K | s1K questions | 60 | 13 | 19 | 8 | 0 |
>
> *Table 2*. Robustness to harder negative controls on Gemma-3-4B-Instruct, layer 22. Instead of contrasting against The Pile, we use corresponding question sets as harder negatives, serving as proxies for math-heavy non-reasoning text. TD = token-driven, PTD = partially token-driven, WTD = weakly token-driven, CD = context-dependent.
>
> ---
>
> We hope these clarifications are helpful. We would be happy to address any further questions or concerns the reviewer may have.

---

> > ### Author Rebuttal · Reviewer_ocLu · 2026-04-01
> >
> > Thank you for your responses and additional experiments on different aggregation method and chunk length.

---

> > > ### Author Response · Authors · 2026-04-03
> > >
> > > Thank you for the update and for taking the time to read our responses and additional experiments carefully. We appreciate your thoughtful feedback throughout the review process, and we are glad that our clarifications addressed your concerns. We are grateful for your consideration of our work.

---

### Official Review · Reviewer_VWey · 2026-03-12

**Soundness:** 3
**Presentation:** 3
**Significance:** 2
**Originality:** 2
**Overall Recommendation:** 4
**Confidence:** 4

**Summary:**

This paper responds to recent works that identify reasoning-specific SAE features by comparing feature activation rates on reasoning versus non-reasoning text datasets. The authors define "genuine reasoning features" as features that are specific to reasoning, not explained by simple textual patterns, and stable under meaning-preserving paraphrases. They then show that no contrastively selected features satisfy these criteria, using n-gram-based token injections and LLM-generated rewrites to test each property. The paper also provides theoretical analysis suggesting that SAE training biases representations towards lower-dimensional correlates, along with preliminary results on feature steering.

**Compliance With Llm Reviewing Policy:**

Affirmed.

**Final Justification:**

Following a productive discussion, I'm bumping to weak accept and increasing my significance and originality ratings. While I still fundamentally view this paper as defining a stronger notion of reasoning features and then falsifying it, I found the authors' discussion of how explicitly articulating such a claim can productively shape future discourse and work in this area compelling. The new dimensionality results are also a nice empirical complement to the theory, which I had initially had some concerns about being not particularly empirically grounded and too stylized.

**Key Questions For Authors:**

Can the authors point to specific prior works that make claims similar to their notion of "genuine reasoning features," either for reasoning or for other abstract properties as discussed in Appendix A? My general impression is that this is a fairly unconventional view — that such strong features might exist in any domain — and clarification here would help me assess the significance of the negative result.

**Limitations:**

yes

**Strengths And Weaknesses:**

**Strengths:**

- The experimental rigor in carefully evaluating identified SAE features is thorough
- The range of models studied is appreciated, covering multiple model families including Gemma, Llama,  R1-distilled model.
- The definition of "genuine reasoning features" is clearly articulated and helpful for understanding the precise claim being falsified.
- The observation that contrastively selected features may latch onto low-dimensional correlates is intuitive and well-motivated.
- The extended discussion in Appendix A is helpful and adds useful nuance to the paper's claims.

**Weaknesses:**

- To my knowledge, the cited works using contrastive methods generally make humble claims — that certain features seem to correspond with reasoning cues and have certain effects — while remaining skeptical and making no strong claim of "genuine reasoning features." This paper is therefore simultaneously defining a stronger notion of reasoning feature and then falsifying it, which somewhat limits the impact of the negative result.
- There are natural ways to apply study reasoning features (e.g., cross-coders, probing) that are not addressed. Although this is discussed and acknowledged, it limits the scope of the paper. If the goal is to assess whether such representations exist, proposing a new method or evaluating several approach for finding reasoning representations would be more compelling than falsifying one specific methodology.
- The theoretical analysis assumes reasoning corresponds to a high-dimensional component, which is not empirically grounded. It is unclear why reasoning should be "higher-dimensional," given that SAEs recover both low-level and high-level semantic features. It would be very interesting if this were in fact the case, and empirical results suggesting so would strengthen the paper significantly — but as it stands, this is taken as a given in the theoretical analysis, which feels unintuitive.

---

> ### Author Rebuttal · Authors · 2026-03-30
>
> We thank Reviewer VWey for the insightful comments.
>
> ---
>
> **W1.** Prior contrastive papers make relatively careful claims, so defining a stronger notion of reasoning feature and then falsifying it may limit the impact.
>
> **A1.** Our goal is not to criticize prior contrastive SAE work, but to clarify what can and cannot be concluded when a feature tracks reasoning data and supports steering. From an interpretability perspective, the key question is whether such a feature captures the reasoning process itself or a simpler predictive correlate. We therefore make explicit a standard that is often left implicit. This matters because recent papers describe SAE latents as features that “drive reasoning,” “capture reasoning abilities,” or are “causally associated” with reasoning behavior [1-3]. These are useful results, but they also show why stronger falsification is needed: otherwise readers may overestimate what sparse features establish about reasoning itself. Our contribution is constructive: we provide a clearer framework for studying high-level behaviors and separating reasoning-related cues from underlying reasoning computation.
>
> ---
>
> **W2.** Other approaches such as cross-coders or probing are not addressed.
>
> **A2.** We agree that methods beyond standard SAEs matter, and we conducted new experiments to test whether our conclusions extend beyond one sparse decomposition method. Cross-coders and cross-layer transcoders are useful for model diffing and attribution analysis, but they still use sparse feature decompositions and thus inherit the same sparsity bias relevant to our theory. Consistent with this, in new layer-17 cross-layer transcoder experiments on Gemma-2-2B, only 10/100 features on s1k and 28/100 features on General Inquiry Thinking are context-dependent after token injection, and all evaluated context-dependent features are classified by our LLM evaluator as not genuine reasoning features. Probing is also interesting, but reasoning behavior and reasoning-related lexical cues are tightly coupled, so a probe can recover a predictive signal without isolating the underlying computation. We therefore view our paper as identifying a broader limitation of current sparse-feature approaches. These new results also motivate a future direction: model reasoning as a unified activation-space manifold and study how to map and steer that manifold directly.
>
> ---
>
> **W3.** The theoretical analysis assumes reasoning is high-dimensional, but this is not empirically grounded.
>
> **A3.** To address this directly, we conducted new activation-space experiments that will be added in the revision. Using reasoning chains from s1k, we annotate sentences into behaviors such as deduction, backtracking, and uncertainty estimation, then measure the intrinsic dimension of their residual activations. At layer 17, pooled reasoning segments have intrinsic dimension 15.37, while several sub-behaviors are similarly or more expansive, including deduction (15.29), backtracking (18.23), and uncertainty estimation (20.29). Using the General Inquiry Thinking dataset, we also compare reasoning-vs-non-reasoning classification to a matched lexical task: detecting *wait* reaches 90% accuracy with only 5 principal components, whereas reasoning-vs-non-reasoning requires 24. Even paraphrases of the same reasoning chain still have intrinsic dimension above 5 on average. Together, these results suggest that reasoning is represented as a distributed family of partially overlapping submanifolds rather than a near-one-dimensional code, which is exactly the assumption required by our theory.
>
> ---
>
> **Q1.** Can the authors point to prior works making claims similar to their notion of “genuine reasoning features”?
>
> **A4.** Our point is not that prior work already adopts our exact definition. Rather, many recent papers use looser operational notions of reasoning, such as CoT traces, CoT-vs-noCoT contrasts, strategy keywords, or reasoning-associated vocabulary [4-6,1]. These are reasonable starting points, but they leave open the core interpretability question: are we isolating the internal reasoning process, or only sparse handles correlated with it and useful for control? Our contribution is to make that target explicit and provide a more rigorous framework for assessing what current sparse-feature methods do and do not establish about reasoning representations.
>
> ---
>
> **References:**
>
> [1] I Have Covered All the Bases Here: Interpreting Reasoning Features in Large Language Models via Sparse Autoencoders
>
> [2] Resa: Transparent Reasoning Models via SAEs
>
> [3] Reasoning Beyond Chain-of-Thought: A Latent Computational Mode in Large Language Models
>
> [4] Feature Extraction and Steering for Enhanced Chain-of-Thought Reasoning in Language Models
>
> [5] How does Chain of Thought Think? Mechanistic Interpretability of Chain-of-Thought Reasoning with Sparse Autoencoding
>
> [6] Controllable LLM Reasoning via Sparse Autoencoder-Based Steering

---

> > ### Author Rebuttal · Reviewer_VWey · 2026-04-04
> >
> > Thank you for the thoughtful discussion — this is really helpful for understanding the paper's contribution and positioning.
> >
> > **W1/W2/Q1:** I appreciate the clarification, and better understand and appreciate the goal and value of such a constructive framing. I will raise my significance and originality scores accordingly. That said, it doesn't fully resolve my concern that the paper is simultaneously defining a stronger notion of reasoning feature and then falsifying it, which limits the impact.
> >
> > **W3:** These new intrinsic dimension results sound really fascinating and have potential to change my assessment of the contribution. Given the limited rebuttal space, I don't fully follow the details — for example, how intrinsic dimension is being measured, or how the dataset is selected between the two settings. Could you share more about the methodology and experimental setup?

---

> > > ### Author Response · Authors · 2026-04-04
> > >
> > > Thank you for the thoughtful follow-up. Your questions are very helpful because they get at both the conceptual target of the paper and the role of the new activation-space evidence.
> > >
> > > On the first point, we do not believe our definition is stronger than needed. From an interpretability perspective, calling something a **“reasoning feature”** suggests more than correlation with reasoning-labeled data. It suggests that the latent reflects the underlying reasoning computation, rather than a surface cue that often co-occurs with it. If a feature mainly flips with a token, phrase, or stylistic marker, it may still be useful for steering, but it is not yet evidence that we have isolated the reasoning process itself. This is why we require not only reasoning specificity, but also resistance to lexical confounds and stability under meaning-preserving rewrites. In that sense, **our definition is not adding an extra burden on top of prior work**. It makes explicit the minimum standard needed to support an interpretability claim about reasoning itself, rather than about a controllable correlate. This becomes especially important if reasoning is distributed rather than sparse, as suggested by recent work [1] showing that some language-model representations are irreducibly multi-dimensional rather than one-dimensional features.
> > >
> > > On the intrinsic-dimension results, the setup is as follows. We use Gemma-3-4B-it and focus on layer 17, which gave the clearest middle-layer signal. To measure dimension, we apply the **maximum-likelihood intrinsic-dimension estimator** of [2] to clouds of hidden states in residual space. Intuitively, this estimates how many local degrees of freedom are needed to explain the geometry of nearby activations. We ran three geometry analyses on annotated reasoning traces from s1K, using Gemini and DeepSeek chains. First, to estimate the size of the reasoning manifold as a whole, we use segments of reasoning chains and estimate intrinsic dimension on the pooled hidden-state cloud. This gave **15.37**. Second, we estimate dimension separately for six behavior labels: initializing, deduction, adding knowledge, example testing, uncertainty estimation, and backtracking. These ranged from 14.83 to 20.29, with deduction at **15.29**, backtracking at **18.23**, and uncertainty estimation at **20.29**. The main point is that the global reasoning cloud is not a tiny one-dimensional code, and several fine-grained reasoning behaviors are equally expansive than pooled reasoning. Third, we study meaning-preserving variation by taking short source passages from the same corpus, generating many paraphrases for each one, and estimating dimension within each paraphrase family. Here the mean intrinsic dimension was **5.17**. So when the reasoning content is intended to stay fixed and only wording changes, the geometry contracts sharply.
> > >
> > > We then added a matched decoding comparison on the General Inquiry Thinking Chain-of-Thought dataset, which contains general-domain reasoning traces and answer text. In this setting, we chunk text, pool activations, reduce them with PCA, and train the same linear classifier in two tasks under the same representation. In one task we classify **reasoning versus non-reasoning** answer text. In the other, we classify **whether the chunk contains the word “wait.”** The lexical baseline is easier at every dimensionality budget: “wait” reaches 90% accuracy with only 5 principal components, whereas reasoning versus non-reasoning needs 24, and the lexical task has higher accuracy at every tested number of principal components. This suggests that **local lexical structure is concentrated in a much smaller subspace than reasoning status**. Taken together, these results strengthen the conceptual point behind our theory: reasoning appears as a broader distributed family of partially overlapping manifolds, while meaning-preserving rewrites occupy a tighter local neighborhood. This makes the theoretical argument more concrete and helps explain why sparse features can recover useful handles on reasoning-related behavior without spanning the reasoning process itself.
> > >
> > > For convenience, the main new numbers are:
> > >
> > > | Setting | Result |
> > > | --- | --- |
> > > | Pooled reasoning manifold | 15.37 |
> > > | Deduction | 15.29 |
> > > | Backtracking | 18.23 |
> > > | Uncertainty estimation | 20.29 |
> > > | Paraphrases of same reasoning passage | 5.17 |
> > > | 90% accuracy for “wait” detection | 5 principal components |
> > > | 90% accuracy for reasoning vs non-reasoning | 24 principal components |
> > >
> > > We will incorporate these clarifications and the new activation-space discussion in the revision. We hope these clarifications are useful, and we would appreciate your consideration in updating the score in light of the additional evidence above.
> > >
> > > ---
> > >
> > > **References:**
> > >
> > > [1] Not All Language Model Features Are One-Dimensionally Linear. ICLR 2025.
> > >
> > > [2] Translated Poisson Mixture Model for Stratification Learning. International Journal of Computer Vision.

---

### Decision · Program_Chairs · 2026-04-30

**Decision:**

Accept (regular)

**Comment:**

This paper questions whether sparse autoencoders recover genuine monosemantic reasoning features, and proposes a falsification framework based on token injection and counterexample generation to test that claim.

Reviewers generally agreed that the paper is well executed and addresses an important question for current mechanistic-interpretability work. The empirical evaluation was seen as broad and careful, and the rebuttal appears to have resolved several initial concerns by adding robustness checks, harder negatives, longer-context analyses, and cross-LLM validation for the falsification step.

At the same time, some reservations remain about the precise scope and impact of the negative result. In particular, reviewers noted that the paper defines a relatively strong notion of "genuine reasoning features" and then falsifies that notion for standard contrastive SAE pipelines, which may limit how broadly the conclusion should be interpreted. There were also some remaining concerns about how much the theoretical analysis adds beyond motivating an intuitive sparsity bias, and about the extent to which the paper points toward concrete alternatives for studying distributed reasoning representations.

Overall, however, the paper seems to make a substantive and timely contribution by sharpening what current sparse-feature methods do and do not establish about reasoning, and by providing a stronger evaluation framework for future work in this area. So while I think this is somewhat borderline, I lean towards acceptance.